# FedAvP: Augment Local Data via Shared Policy in Federated Learning

**Minui Hong**[*]    **Junhyeog Yun**[*]    **Insu Jeon**[†]    **Gunhee Kim**[*]

Seoul National University, Seoul, South Korea

[*]{alsdml123,junhyeog,gunhee}@snu.ac.kr

[†]{insuj3on}@gmail.com

## Abstract

Federated Learning (FL) allows multiple clients to collaboratively train models without directly sharing their private data. While various data augmentation techniques have been actively studied in the FL environment, most of these methods share input-level or feature-level data information over communication, posing potential privacy leakage. In response to this challenge, we introduce a federated data augmentation algorithm named FedAvP that shares only the augmentation policies, not the data-related information. For data security and efficient policy search, we interpret the policy loss as a meta update loss in standard FL algorithms and utilize the first-order gradient information to further enhance privacy and reduce communication costs. Moreover, we propose a meta-learning method to search for adaptive personalized policies tailored to heterogeneous clients. Our approach outperforms existing best performing augmentation policy search methods and federated data augmentation methods, in the benchmarks for heterogeneous FL.

## 1   Introduction

Federated Learning (FL) is a collaborative learning approach that allows multiple clients to learn without sharing their private information [1–6]. A central server coordinates the training process across multiple devices and aggregates the locally trained models into a global one; thus reducing the communication cost of exchanging raw data and mitigating the risk of privacy leakage associated with data sharing [7].

However, the limited accessibility of data in FL still poses many challenges, such as insufficient training data and local data bias. To address these challenges, there has been a growing interest in federated data augmentation techniques [8–12]. They aim to increase the diversity and volume of data available at each client, thereby improving the overall robustness and performance of the federated models. For example, FedMix [8] improves performance and privacy by averaging multiple images to facilitate data mixup among clients. Similarly, FedFA [11] utilizes feature statistics to mitigate local data biases, to improve model generalization. Despite their benefits, these methods often apply the sharing of input-level [8, 10, 12] or feature-level [13, 11] information. Such information sharing poses additional privacy concerns since malicious attackers could potentially reconstruct original data by applying gradient matching loss [14] on the additional input and feature information.

We propose a novel federated data augmentation algorithm named FedAvP (**A**ugmet Local Data **v**ia Shared **P**olicy in Federated Learning), which shares only the *augmentation policies* during training. Thus, each client does not need to share its own data or data-related information directly but obtains collective knowledge on how to augment the dataset at local learning. Previously, AutoAugment [15] utilizes reinforcement learning (RL) to automatically find the optimal data augmentation policy for a target dataset. While AutoAugment requires extensive GPU resources, more efficient and faster policy search has been studied [16–20]. However, these methods are not designed for FL environments but

they perform policy search for public batch datasets. The data scarcity and heterogeneity in FL could fail these standard policy search frameworks.

To improve the robustness and generalization of model training in the heterogeneous FL setting, we first introduce a Federated Meta-Policy Loss (FMPL) specifically designed to compute a gradient of augmentation policy that updates a shared data augmentation policy for each client's unique environment. Our approach guides the policy gradient to account for the effects of data augmentation on the unseen local data. The policy gradient utilizes higher-order information; the impact of the data augmentation is estimated by its effect on the validation loss observed after a few gradient descent steps with the initial augmented data. However, computing the direct meta-policy gradient in FL requires an additional communication step between the server and clients. To bypass this, we also develop an alternative meta-policy search method that utilizes a first-order approximation. We further demonstrate that the adaptive policy search technique can adapt to heterogeneous data distributions among clients in the FL environment.

In the experiments, our FedAvP demonstrates superior performance on CIFAR-10/100 [21], SVHN [22], and FEMNIST [23] datasets within an FL context, compared to existing federated learning algorithms, including FedAvg [2], FedProx [4], FedDyn [5], FedExP [6], and federated data augmentation algorithms, including FedGen [9], FedMix [8], and FedFA [11]. Moreover, to further leverage the potential performance of these algorithms, we also conducted experiments applying data augmentation techniques such as RandAugment [17] and TrivialAugment [24] to these algorithms for comparison. We also evaluate our algorithm in environments where data is non-i.i.d. with heterogeneous clients [25]. We further compare the effectiveness of the utility of sharing this policy across clients for searching, in contrast to conducting a local policy search.

Our primary contributions are as follows.

1. We propose FedAvP(Augment Local Data via Shared Policy in Federated Learning) as the first algorithm in federated learning that facilitates shared augmentation policies among clients for federated policy search, to the best of our knowledge.

2. We introduce the federated meta-policy loss for effective policy search, and further propose a first-order approximation to this loss to enhance privacy and reduce communication costs.

3. Enabling meta-learning, our algorithm allows for rapid adaptation of a personalized policy by each client, addressing the challenge of highly heterogeneous data distributions among clients in the FL environment.

## 2 Related Work

**Automated Data Augmentation**. AutoAugment [15] utilizes RL to find an optimal data augmentation policy for a target dataset automatically. FastAA [16] proposes a more efficient search strategy by training small NNs in parallel without iterative training, using the density matching method. RandAugment [17] suggests a simplified search method composed of two hyper-parameters, which find the augmentation policy without a separate search process. TrivialAugment [24] further simplifies the algorithm and applies a single augmentation to each image as a parameter-free method. MetaAugment [19] proposes a sample-aware augmentation policy network to capture the variability of training samples more accurately than previous dataset-based search methods. Deep AutoAugment [18] proposes a fully automated search method that builds a multi-layer data augmentation pipeline from scratch by stacking augmentation layers. All of these recent data augmentation methods were developed under the assumption that all training data is accessible on a server. This assumption is invalid in FL since data privacy is a significant concern and it is challenging to tune these algorithms for each of the numerous heterogeneous local datasets. Therefore, our study aims to develop a new data augmentation policy search algorithm that takes into account the distributed FL process while preserving data security. RandAugment [17] and TrivialAugment [24] can be applied simultaneously to many clients in a federated learning environment, so we compared these methods in our experiments.

**Federated Data Augmentation**. The standard Federated Learning (FL) framework, such as FedAvg [2], typically performs iterative local model updates at each client and a global update at a server. Since the clients and server only communicate through the model parameters instead of raw data, it enables secured and decentralized learning [1, 3]. Despite its benefits, FL still has challenges

such as convergence degradation and model overfitting due to heterogeneity and sparsity among the data caused by the differences in client's actions and preferences [25]. To address these challenges, federated data augmentation (FDA) employs a data augmentation approach instead of a model-centric approach. FedMix [8] applies Mixup [26] to FL, augmenting data by linear interpolation of two random training examples and their labels. For privacy reasons, FedMix transfers mixup data to the server by averaging multiple images from the local device. In FedGen [9], the server learns a lightweight generator to ensemble user information, which is then broadcasted to users to regulate local training using the learned knowledge. FedFA [11] assumes that the data distribution of the clients can be summarized by the statistics of the latent features (*i.e.*, mean and standard deviation). This allows learning local models by regularizing the gradients of the latent representations, weighted by the variances of the feature statistics estimated from the entire client federation. StatMix [13] sends the mean and standard deviation information of local client images to the server and makes it available for learning for each client. These methods use input-level data (image) averaging or feature-level statistics to prevent direct data transfer. Unlike previous methods, our methodology focuses on transmitting only the policy information optimized for local datasets from each client. ATSPriavacy [27] has demonstrated that searching for transformation policies can also protect against reconstruction attacks in Federated Learning (FL), while preserving performance. We compare our algorithm with this approach in terms of both vulnerability to reconstruction attacks and performance in the experimental section

## 3 Approach: FedAvP

We introduce FedAvP (Augment Local Data via Shared Policy in Federated Learning), which performs data augmentation search by sharing policies among clients in a federated learning (FL) environment. Starting from the problem formulation (§3.1), we address the challenges of heterogeneous clients with a proposal for adaptive policy search (§3.2). Finally, we extend its applicability through integration with the FedAvg algorithm [1] and joint learning (§3.3).

### 3.1 Problem Formulation

Our approach is based on the idea of centralized FL [7]; it is not possible to share personal data neither between the server and clients nor among clients. This is different from traditional automated policy search algorithms, which find the optimal policy in a single batch dataset [15, 16, 20, 19, 18].

**Objective**. We begin with the standard FL algorithm, FedAvg [1], aiming to find augmentation policies that minimize a given objective function as follows:

$$\text{Model: } \min_{w} \sum_{k=1}^{K} \alpha_k \ell \left( w; t_{p_\theta}(D_k^{\text{train}}) \right), \quad \text{Policy: } \min_{p_\theta} \sum_{k=1}^{K} \alpha_k \ell \left( w; D_k^{\text{val}} \right). \tag{1}$$

$K$ is the number of clients used to train the global model $w$, and $D_k$ is the local data of client $k$. The transformation policy $p_\theta$ is used to augment the data denoted by $t_{p_\theta}(D_k^{\text{train}})$ with coefficients $\alpha_k$ satisfying $\sum \alpha_k = 1$ and $\alpha_k \geq 0$. Assuming client $k$ has $n_k$ data samples, $\alpha_k$ is defined as $\alpha_k = \frac{n_k}{n}$ where $n = \sum_k n_k$. Under a global policy assumption, all clients share the same $p_\theta$. Alternatively, if we assume that each client has its own transformation policy $p_{\theta^k}^{\text{local}}$, we represent $p_\theta^{\text{local}} = \{p_{\theta^1}^{\text{local}}, p_{\theta^2}^{\text{local}}, \dots, p_{\theta^K}^{\text{local}}\}$.

**Search Space**. We use the augmentation space having a sequence of two operations, following the search space from previous studies on automated policy search [17, 19]. We examine a set of 17 operations in total, including {Identity, ShearX/Y, TranslateX/Y, Rotate, AutoContrast, Equalize, Solarize, Posterize, Contrast, Color, Brightness, Sharpness, RandFlip, RandCutout, RandCrop}. Each operation includes a random magnitude, rescaled and uniformly sampled from the normalized interval $[0, 1]$. Assuming each of the $K$ clients has individual policy, the entire search space consists of $K$ joint distributions, each having a size of $17 \times 17$. For a global policy, we learn a single $17 \times 17$ dimensional joint distribution corresponding to the two operations. Specifically, $p_\theta$ has a value in $[0, 1]$ using the sigmoid function on $\theta$. For augmentation sampling, we normalize the policy parameter vector $p_\theta$ whose sum to be one, and sample from a joint distribution with a regularization term $\epsilon$. Operation pairs $(op1, op2)$ are drawn from a mixed distribution:

$$(op1, op2) \sim (1 - \epsilon) \cdot \frac{p_\theta}{\sum(p_\theta)} + \epsilon \cdot \frac{1}{17^2} \tag{2}$$

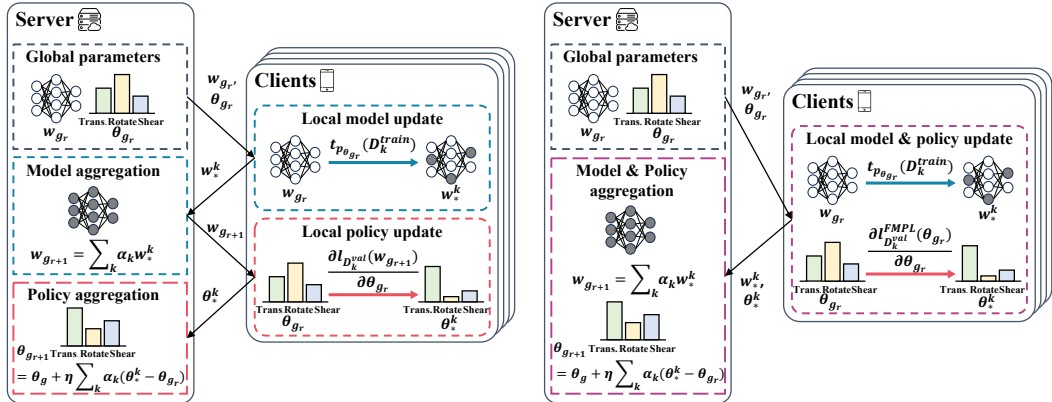

(a) Federated policy optimization using our FMPL

(b) First-order approximation of federated policy optimization

Figure 1: Overview of FedAvP. (a) The server sends global model parameters $w_{g_r}$ and policy parameters $\theta_{g_r}$ to clients. Clients train local models with augmented data, and the server aggregates them to compute $w_{g_{r+1}}$. Clients update policies on $w_{g_{r+1}}$ using validation data, and the server aggregates these policies. (b) Clients update the model and policy parameters via first-order approximation. The server aggregates client updates to form the updated global model $w_{g_{r+1}}$ and policy parameters $\theta_{g_{r+1}}$.

This sampling strategy utilizes a uniform probability across all operation pairs for a balanced exploration and exploitation, as adopted in previous work [19].

## 3.2 Policy Optimization

To search the policy in a differentiable manner, we adopt the concept from meta-learning [28], where a model is trained on training data $n$ times by SGD algorithm ($n$ inner steps), followed by validation of one outer step. In our context, training is performed on augmented data, followed by evaluation on validation data [29, 19]. However, directly applying this concept of inner and outer steps to FL is quite challenging due to the added complexity of FL, which involves training local models and aggregating them to update the global model.

Since our goal is to optimize the global augmentation policy, we redefine the inner and outer steps from the FL perspective. In a standard FL setting, the server sends a global model to the clients for each round, which is trained with their local training data. Afterward, the aggregation process occurs on the server to update the global model. We can regard one round of local training and aggregation as *one inner step*. After $r$ rounds, where the global model is updated $r$ times, validating the final updated model on each client can be considered *one outer step*. We set $r = 1$, meaning validation occurs after each round.

**Federated Meta-Policy Loss**. In a single round of client updates, we first perform local training on each client and aggregate the local models to update the global weights, which are then redistributed to the clients for computing the validation loss. Figure 1 (a) illustrates this process. In each round, the initial weight for a client $k$, denoted $w_0^k$, is set to the global weight $w_{g_r}$. The local training consists of $N$ iterations, with a batch size $B$ of the training data $D_{k,n}^{\text{train}}$ at each iteration $n$. A transformation according to the policy $p_{\theta_n^k}$ is applied to each data sample. The local loss at iteration $n$ for client $k$ is calculated as

$$\text{Local Loss} = \frac{1}{B} \sum_{i=1}^{B} P_{\theta_n^k}^i \ell(w_n^k; t_{p_{\theta_n^k}}^i(D_{k,n}^{\text{train}(i)})), \tag{3}$$

where $P_{\theta_n^k}^i$ is the unnormalized probability $p_{\theta_n^k}(op1, op2)$ for the $i$-th transformation, when the sampled transformation $t_{p_{\theta_n^k}}^i$ is the operations $(op1, op2)$. This reweighting strategy is inspired by recent sample reweighting [29, 19]. Following the local updates with the augmentation policy $p_{\theta^k}$, we aggregate the results to obtain the new global weights $w_{g_{r+1}}$, which are redistributed to the clients for validation loss assessment. At the start of round $r$, after distributing the global weights $w_{g_r}$ to

each participating client, the procedure for the federated meta-policy loss can be summarized as follows:

1. Each client $k$ performs local updates by optimizing their local weights $w_n^k$ using the augmented data with the local loss in Eq. (3). The updated local weights are then aggregated according to $w_{g_{r+1}} = \sum_k \alpha_k w_N^k$.

2. The aggregated global weights $w_{g_{r+1}}$ are then sent back to the same clients.

3. The Federated Meta-Policy Loss (FMPL) is computed on each client's validation set.

$$\text{FMPL} = \ell_{D_k^{\text{val}}}(w_{g_{r+1}}). \tag{4}$$

For this loss, the gradient with respect to the augmentation policy $p_\theta$ is computed using backpropagation. Nonetheless, this approach presents two significant challenges. Firstly, it necessitates access to validation gradient information of other clients, which poses privacy concerns. Secondly, it requires revisiting the same clients for additional updates, thereby doubling the communication overhead.

**First-order Approximation**. To ensure security by preventing access to other clients' gradients and to reduce communication costs, we derive an approximation for the policy gradient with respect to $\theta_{n-1}^k$ at the local step $n$ of the client $k$ by a Taylor expansion, as in the following proposition. See Appendix B for more details.

**Proposition 1.** *Consider the federated meta-policy loss derived from the updated weight $w_n^k$ for client $k$ at step $n$ using a first-order Taylor expansion:*

$$\ell_{D_k^{\text{val}}}(w_{g_{r+1}}) \approx \ell_{D_k^{\text{val}}}(w_n^k) + \nabla \ell_{D_k^{\text{val}}}(w_n^k)^T (w_{g_r} - w_n^k). \tag{5}$$

*When computing the policy gradient of the loss with respect to $\theta_{n-1}^k$, the first-order gradient approximation is*

$$-\alpha_k \cdot lr \frac{\partial (\nabla \ell_{D_k^{\text{val}}}(w_n^k)^T \nabla \ell_{t_{p_{\theta_{n-1}^k}}}(D_{k,n-1}^{\text{train}})(w_{n-1}^k))}{\partial \theta_{n-1}^k}, \tag{6}$$

*where $w_n^k = w_{g_r} - lr \cdot g_{w_0^k}^{aug} - \ldots - lr \cdot g_{w_{n-1}^k}^{aug}$ and $\alpha_k$ is a coefficient proportional to the client's data size.*

In Proposition 1, $g_{w_n^k}^{aug} = \nabla \ell_{t_{p_{\theta_n^k}}(D_{k,n}^{\text{train}})}(w_n^k)$ is the gradient obtained from the local loss in Eq. (3) at step $n$ for client $k$. We calculate the gradient within the same client to prevent gradient leakage across clients. We optimize a policy that maximizes the inner product of the gradient obtained from sampling the validation data and the gradient obtained through augmentation using $p_{\theta_{n-1}^k}$. The validation data are not used separately; instead, they are replaced by sampling the next batch, inspired by Reptile [30]. Proposition 1 also indicates that policy gradients can be computed concurrently with local updates. We utilize this approach to joint training (§3.3), which enables simultaneous model and policy training.

**Gradient Clipping**. From the policy gradient of Eq. (6), it becomes evident that policy search aims to maximize the dot product of the gradients derived from both augmentation and validation data. This approach, however, can introduce a bias toward augmentations with larger gradient magnitudes due to the nature of the dot product. Inspired by [31, 18], we mitigate the influence of gradient magnitude by gradient clipping [32, 33]. We apply gradient clipping to the gradients from both validation and augmentation data in Eq. (6) using a regularizer hyperparameter $c$.

**Adaptive Policy Search**. Our first-order approximation computes policy gradients at each local update step $n$ without aggregation. These gradients are then averaged across all steps for each client to update the policy after a single communication round. However, this method can slow the policy search, since it only permits one gradient descent update per communication round. Drawing inspiration from meta-learning [28, 30, 34], which quickly adapts to various tasks using neural network training, we propose an adaptive policy search. The augmentation policy is represented as a vector parameter $p_\theta = \text{sigmoid}([\theta_1, \theta_2, \ldots, \theta_{289}])$, where $[\theta_1, \theta_2, \ldots, \theta_{289}]$ denotes an $17 \times 17$ joint distribution. We use a neural network comprising the following dense layers: one dummy embedding layer, two 100-dim hidden layers, and an output layer shaped to the $17 \times 17$ distribution

**Algorithm 1** FedAvP: Joint Training

---

**Input:** # of communication round $R$, # of client $N$, server policy learning rate $\eta$, client model learning rate $\gamma$, client policy learning rate $\lambda$, local steps $E$, gradient clipping threshold $c$, regularization term $\epsilon$.

Initialize the global model parameter $w_{g_r}$ and the global policy parameter $\theta_{g_r}$

**for** $r = 1, ..., R$ **do**
    Sample $K$ clients from $1, ..., N$ clients
    **for** $k = 1, ..., K$ **do**
        Set $w_0^k = w_{g_r}$ and $\theta_0^k = \theta_{g_r}$
        $w_*^k, \theta_*^k \leftarrow$ LOCALUPDATE$(w_0^k, \theta_0^k)$
    $w_{g_{r+1}} \leftarrow \Sigma_k \alpha_k w_*^k$
    $\Delta\theta \leftarrow \Sigma_k \alpha_k (\theta_*^k - \theta_{g_r})$
    $\theta_{g_{r+1}} \leftarrow \theta_{g_r} + \eta\Delta\theta$

**procedure** LOCALUPDATE$(w, \theta)$
    Initialize $w_0^k = w, \theta_0^k = \theta$
    **for** Each local step $n$ from 0 to $E$ **do**
        Sample transformations $t_{p_{\theta_n}}$ from $p_{\theta_n}$
        Sample batch data $t_{p_{\theta_n}}(D_{k,n}^{\text{train}})$ from $D_k$ and $t_{p_{\theta_n}}$
        Compute $\nabla\ell_{t_{p_{\theta_n}}(D_{k,n}^{\text{train}})}(w_n^k)$ using Eq.(3)
        $w_{n+1}^k \leftarrow w_n^k - \gamma\nabla\ell_{t_{p_{\theta_n}}(D_{k,n}^{\text{train}})}(w_n^k)$
        Sample batch data $D_{k,n}^{\text{val}}$ from $D_k$
        Compute $\nabla\ell_{D_{k,n}^{\text{val}}}(w_{n+1}^k)$ at $w_{n+1}^k$
        $\theta_{n+1}^k \leftarrow \theta_n^k - \lambda\nabla\ell_{D_{k,n}^{\text{val}}}^{\text{FMPL}}(\theta_n^k)$ using Proposition 1
    Set $w_*^k = w_{n+1}^k$ and $\theta_*^k = \theta_{n+1}^k$
    Send $w_*^k$ and $\theta_*^k$ to the server

---

size. We update our policy as done in Reptile [30, 35]. That is, the local policy updates on each client correspond to the inner steps in the Reptile, while the global policy updates on the server are analogous to the outer steps, as detailed in Algorithm 1. We train the policy neural network by increasing the dot-product between policy gradients on each client as follows:

$$\theta_{g_{r+1}} \approx \theta_{g_r} - \eta\lambda\frac{\partial}{\partial\theta_0^k}\mathbb{E}\left[\sum_{j=0}^{n} L_{k,j} - \frac{\lambda}{2}\sum_{j=0}^{n}\sum_{s=0}^{j-1}\langle\nabla L_{k,j} \cdot \nabla L_{k,s}\rangle\right], \tag{7}$$

where $L_{k,j} = \ell_{D_{k,j}^{\text{val}}}^{\text{FMPL}}(\theta_0^k)$ is the federated meta-policy loss in Eq. (4) computed on the client $k$'s $j$-th validation data batch using the global policy parameters $\theta_0^k$. $\langle\nabla L_{k,j} \cdot \nabla L_{k,s}\rangle$ is the dot-product between policy gradients on the client $k$. See Appendix C for more details. This process enables the policy neural network to learn in a direction that enhances the dot-product between the policy gradients of each client, thereby facilitating efficient policy search and enabling personalized policy search. We incorporate this strategy in the joint training in §3.3. An experiment result regarding this will be conducted in §4.

### 3.3 Joint Training

As done in AutoAugment, one could employ a pretrained network to perform policy search [16, 18]. However, such methods require a separate training phase and present complications in FL environments, since training a pretrained network beforehand is cumbersome, and the separate phase is disadvantageous for parallel training. Note that we previously adopt an adaptive policy search that can simultaneously train the model and policy search. At each local update step $n$, the weights $w_{n+1}^k$ and gradients concurrently update the model and policy by comparing the gradient on validation data at $w_{n+1}^k$.

Algorithm 1 illustrates this joint training of our FedAvP. It begins by initializing the global model parameter $w_{g_r}$ and the policy parameter $\theta_{g_r}$, which are then sent to the clients from the server. Each client generates augmented data $t_{p_{\theta_n}}(D_{k,n}^{\text{train}})$ using the policy parameter $p_{\theta_n}$ at every step $n$ and updates the model accordingly. Subsequently, the gradient $\nabla\ell_{D_{k,n}^{\text{val}}}(w_{n+1}^k)$ is computed using the newly updated $w_{n+1}^k$. Following Proposition 1, the policy parameter $\theta_n^k$ is updated to maximize the gradient on both validation and augmentation data $\nabla\ell_{t_{p_{\theta_n}}(D_{k,n}^{\text{train}})}(w_n^k)$. This process of model and policy updates is repeated in every local update. Afterwards, the model and policy parameters from each client are aggregated at the server using $\alpha_k$.

**Fast Update** One limitation of joint learning is that it requires backpropagation for the policy gradient at every step. To reduce the computation load on local clients, the policy can be updated periodically instead of at every local step $n$, specifically when $n \mod \tau == 0$. In all our experiments, we set $\tau = 5$. See the variant of the algorithm in Appendix A.3. We also reduced the hidden size of the policy neural network to two 25-dimensional hidden layers. In our experiments in §4, we will compare the performance of the Fast Update.

| Dataset | CIFAR-100 | | CIFAR-10 | SVHN | FEMNIST |
| | $\alpha = 5.0$ | $\alpha = 0.1$ | $\alpha = 5.0$ | $\alpha = 0.1$ | |
| Method | Test (%) | Test (%) | Test (%) | Test (%) | Test (%) |
|---|---|---|---|---|---|
| FedAvg + Default | 40.05 | 37.34 | 79.76 | 85.58 | 80.65 |
| + RandAugment | 47.29 | 43.60 | 82.82 | 84.84 | 79.40 |
| + TrivialAugment | 46.61 | 42.16 | 82.00 | 83.36 | 79.01 |
| FedProx + Default | 40.57 | 37.71 | 80.64 | 86.79 | 81.45 |
| + RandAugment | 45.97 | 41.39 | 82.56 | 85.52 | 77.11 |
| + TrivialAugment | 46.61 | 41.81 | 81.83 | 84.11 | 79.67 |
| FedDyn + Default | 42.09 | 38.52 | 80.36 | 87.60 | 80.47 |
| + RandAugment | 45.70 | 42.24 | 82.51 | 81.47 | 77.64 |
| + TrivialAugment | 46.83 | 41.10 | 82.03 | 83.41 | 79.31 |
| FedExP + Default | 42.76 | 38.28 | 80.64 | 86.66 | 81.45 |
| + RandAugment | 46.13 | 42.23 | 82.86 | 84.63 | 79.69 |
| + TrivialAugment | 48.55 | 42.09 | 82.51 | 83.72 | 80.20 |
| FedGen + Default | 42.14 | 38.27 | 80.23 | 86.79 | 81.86 |
| + RandAugment | 47.11 | 43.10 | 81.90 | 84.39 | 79.34 |
| + TrivialAugment | 47.71 | 40.76 | 82.58 | 83.23 | 77.35 |
| FedMix + Default | 40.26 | 38.69 | 80.99 | 86.02 | 81.63 |
| + RandAugment | 46.69 | 43.00 | 83.08 | 83.44 | 79.46 |
| + TrivialAugment | 46.64 | 42.63 | 81.83 | 82.34 | 77.84 |
| FedFA + Default | 43.70 | 41.21 | 82.61 | 87.33 | 81.13 |
| + RandAugment | 48.86 | 43.44 | 82.44 | 81.32 | 78.71 |
| + TrivialAugment | 47.86 | 43.45 | 80.12 | 78.62 | 78.96 |
| FedAvP (W/ Local Policy) | 49.04 | 43.86 | 83.64 | 87.05 | 83.94 |
| FedAvP (Fast Update) | 49.97 ($\pm$0.04) | 45.08 ($\pm$0.01) | 83.55 ($\pm$0.06) | 87.86 ($\pm$1.53) | 84.47 ($\pm$0.006) |
| FedAvP | 50.47 ($\pm$0.03) | 45.96 ($\pm$0.01) | 83.78 ($\pm$0.004) | 89.81 ($\pm$1.55) | 84.27 ($\pm$0.07) |

Table 1: Classification accuracies with different heterogeneity degrees ($\alpha = 5.0$ and $\alpha = 0.1$) across CIFAR-100/10, SVHN, and FEMNIST datasets. We report results averaged over 3 random seeds with variances for FedAvP (Fast Update) and FedAvP.

## 4 Experiments

### 4.1 Experimental setup

**Environments**. Previous FL studies have demonstrated that the standard algorithms are effective and converged when the data is i.i.d. [36–38]. To evaluate robust performance in non-i.i.d. data, we set up our experimental environment by distributing the CIFAR-10/100 [21] and SVHN [22] datasets with different levels of data heterogeneity among clients. We assign the data to 130 clients based on a Dirichlet distribution with different hyperparameters of $\alpha = [5.0, 0.1]$, as done in pFL-Bench [25]. The smaller $\alpha$ is, the higher the degree of heterogeneity is. Among these clients, only 100 randomly selected clients participate in the training, while the remaining 30 are nominated as out-of-distribution (OOD) clients. In each communication round, only 10 clients are sampled. We employed a standard CNN model, consistent with those in previous studies [39–41], for the global model. Experiments involving the larger model and OOD clients are provided in Appendix A.2. For the FEMNIST dataset [23], we introduced variability in data size by distributing data based on the writers [8, 11]. Further environments details are provided in Appendix A. Our code is available at https://github.com/alsdml/FedAvP.

**Baselines**. For a comprehensive evaluation, we compared our method with state-of-the-art federated learning algorithms such as FedAvg [2], FedProx [4], FedDyn [5], FedExP [6], and federated data augmentation algorithms including FedGen [9], FedMix [8], and FedFA [11]. To further explore the potential performance of these algorithms, we conducted experiments applying data augmentation techniques such as RandAugment [17] and TrivialAugment [24]. We also compared the results with those using default augmentations (random crops and horizontal flipping). Additionally, for comparison with our proposed model, we included FedAvP (W/ Local Policy), which trains each

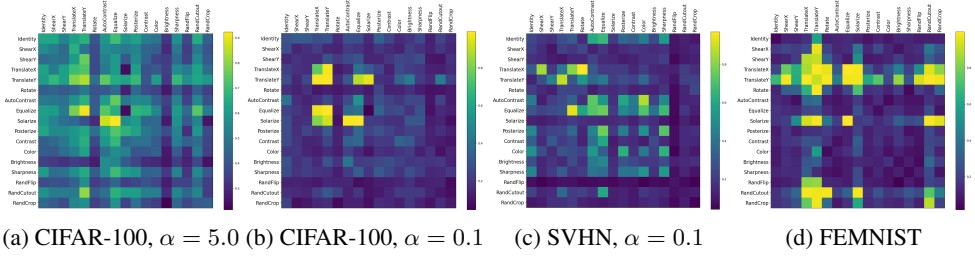

(a) CIFAR-100, $\alpha = 5.0$ (b) CIFAR-100, $\alpha = 0.1$ (c) SVHN, $\alpha = 0.1$ (d) FEMNIST

Figure 2: Visualization of global policies learned in CIFAR-100, SVHN and FEMNIST.

local client without policy aggregation, and FedAvP (Fast Update) which reduces computation load on local clients (Algorithm 2). Further details of baselines are provided in Appendix A.

## 4.2 Performance on Non-i.i.d. Settings

**Settings.** Table 1 reports the *test accuracy*, which measures the accuracy on the test dataset of the 100 participating clients. The test accuracy is calculated as the weighted average of each client's accuracy by the number of data points they have. We compared each baseline with three data augmentations: +Default (random crops and horizontal flipping), +RandAugment, and +TrivialAugment. For FedAvP (W/ Local Policy), each client has its own transformation policy and policy aggregation is removed from our algorithm (Algorithm 1). For FedAvP (Fast Update), we used periodic local updates (Algorithm 2). For FedAvP, we used full local updates (Algorithm 1).

**Results.** The application of automated data augmentation algorithms such as RandAugment and TrivialAugment within a federated learning framework does not consistently enhance performance across all cases. In contrast, our algorithm learned distinct augmentations for each dataset, as depicted in Figure 2. When compared to FedAvP (W/ Local Policy) and FedAvP, notable performance improvements were observed in highly non-i.i.d. scenarios such as $\alpha = 0.1$ in CIFAR-100 and SVHN. For instance, the standard deviation of local dataset sizes in CIFAR-100 with $\alpha = 5.0$ was relatively small at 19.43, while it was significantly higher at 118.00 for CIFAR-100 and 533.60 for SVHN with $\alpha = 0.1$, indicating that imbalanced datasets in non-i.i.d. settings pose challenges for training local policies with smaller local datasets. Although FedAvP (Fast Update) experienced some performance declines compared to FedAvP, it generally achieved higher performance than baseline methods across most datasets.

## 4.3 Policy Adaptation on Clients

In our FedAvP, at the beginning of each round, the participating clients receive a global policy from the server, which is then optimized into a local policy using each client's local data. That is, this optimization adapts the global policy into a personalized policy on the local data of each client. The clients use the personalized policy to train their local model to achieve high performance and to aggregate well with other local models in the server. Figure 3 shows the statistics on the Euclidean distances between

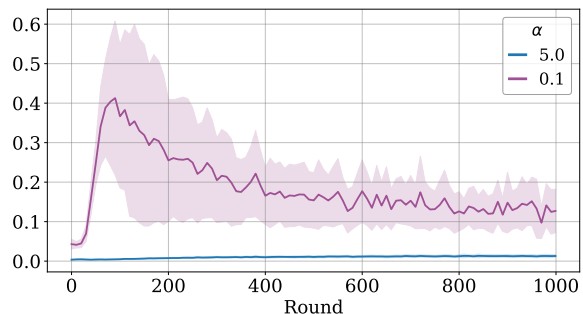

Figure 3: Statistics of personalized policies between different clients on CIFAR-100.

the personalized policies of clients participating in each round and the global policy for that round. With $\alpha = 5.0$ in CIFAR-100, the data is sufficiently i.i.d., and thus the personalized policies of clients tend not to deviate from the global policy. On the other hand, with $\alpha = 0.1$, the deviation from the global policy is initially high but decreases as training progresses, particularly after about 100 rounds. The variance of the Euclidean distances also follows this pattern.

## 4.4 Reconstruction Attack

**Settings.** In collaborative learning systems, it has been reported that gradient leakage attacks can occur [14, 42, 43], leveraging gradients to reconstruct the original training data. We conducted these reconstruction attack [14] experiments to evaluate whether our algorithm, which shares policies in a federated learning setting, provides enhanced privacy compared to FedGen, FedMix, and FedFA. Specifically, FedGen shares information at the generator and label distribution levels, FedMix shares at the input level, and FedFA shares at the feature level.

We included a defense algorithm in the performance comparison, ATSPrivacy [27]. Our experiments were conducted on CIFAR-100 with $\alpha = 0.1$, involving two clients: Client(L), which had the most training data (895 data points), and Client(S), with the least (156 data points). For ATSPrivacy, we used policies from [27] that showed the highest accuracy performance. Further experimental details are provided in Appendix A.1.

**Results.** The experimental results for the Reconstruction Attack are summarized in Table 2. Lower PSNR values indicate that the reconstructed images are less similar to the original data, thereby reflecting better privacy preservation. Fed-

| Metric | PSNR | | Accuracy |
|---|---|---|---|
| Method | Client(S) | Client(L) | Test(%) |
| FedAvg | 10.88 | 11.36 | 37.34 |
| FedGen | 8.86 | 9.27 | 38.27 |
| FedMix | 10.27 | 10.48 | 38.69 |
| FedFA | 10.86 | 11.82 | 41.21 |
| **FedAvP** | 8.72 | 9.25 | **45.96** |
| FedGen + label + generator | 9.21 | 9.81 | 38.27 |
| FedMix + input | 11.89 | 12.40 | 38.69 |
| FedFA + feature | 12.11 | 12.87 | 41.21 |
| **FedAvP + policy gradients** | 8.77 | 9.20 | **45.96** |
| ATSPrivacy (7-4-15) | 8.45 | 8.89 | 38.61 |
| ATSPrivacy (21-13-3,7-4-15) | **6.70** | **6.69** | 36.42 |

Table 2: Reconstruction Attack Results

Gen+label+generator utilizes the label distribution of local data and the generative model. FedMix+input and FedFA+feature represent the results of attacks utilizing input level and feature level information, respectively. FedAvP+policy gradients denotes the results of attacks using the policy gradient of our algorithm. ATSPrivacy recorded the lowest PSNR values, indicating that it makes reconstruction difficult. However, this was coupled with a decline in accuracy performance. Despite increases in PSNR for FedGen, FedMix and FedFA, our algorithm's use of policy gradients did not elevate PSNR values.

## 4.5 Computation and Communication Cost

| | CIFAR-100 dataset | |
|---|---|---|
| Method | Rounds(35%) | Time(35%) |
| FedAvg + Default | 300 | 1.05 hours |
| FedAvg + RandAugment | 300 | 1.62 hours |
| FedAvg + TrivialAugment | 450 | 2.17 hours |
| FedAvP (Fast Update) | 200 | 1.18 hours |
| FedAvP | 200 | 4.01 hours |

Table 3: Computation Time in CIFAR-100

| | CIFAR-100 dataset | |
|---|---|---|
| Method | Before(MB) | Per round(MB) |
| FedAvg | 0.00 | 15.35 |
| FedMix | 1.27 | 15.35 |
| FedFA | 0.00 | 15.38 |
| FedAvP (Fast Update) | 0.00 | 15.73 |
| FedAvP | 0.00 | 17.47 |

Table 4: Communication costs in CIFAR-100

For computational comparison, we measured the time taken to reach a target accuracy of 35% on the CIFAR-100 dataset with $\alpha = 0.1$. Regarding communication comparison in FedMix, the method involves sending an average image to the server prior to training, which is detailed in the "Before" section of the table. For FedAvP (Fast Update) using small neural networks, although it is higher than these baselines, the increase of 0.38MB from the cost of FedAvg is about 2.48% compared to the gradient transmission cost of the model. See Appendix A.4 for additional results.

## 5  Conclusion

We proposed a novel federated data augmentation method named FedAvP (Augment Local Data via Shared Policy in Federated Learning). It shares the augmentation policies during training rather than preprocessed or encoded data such as the average of data or statistics of features. Direct exposure of personal information was constrained, yet clients still benefited from the policies learned and shared

across the clients to augment their local data. We also proposed Federated Meta-Policy Loss (FMPL) and used the first-order gradient information to enhance privacy with reduced communication costs. A potential limitation of our algorithm is the introduction of Joint Training. This approach requires consideration when applying our federated data augmentation method to existing federated learning algorithms. Also, our algorithm assumed the use of FedAvg, which does not account for model personalization [44, 45] in the computation of FMPL. Investigating a policy loss that aligns with model personalization algorithms would be interesting.

## Acknowledgment

This work was partially supported by Institute of Information & communications Technology Planning & Evaluation (IITP) grant funded by the Korea government (MSIT) (No. RS-2021-II211343, Artificial Intelligence Graduate School Program (Seoul National University)), by Samsung Advanced Institute of Technology, and Center for Applied Research in Artificial Intelligence(CARAI) grant funded by Defense Acquisition Program Administration(DAPA) and Agency for Defense Development(ADD) (UD230017TD)

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

# A  Implementation Details

**Model**    Consistent with prior studies, we employed a standard CNN model for all experiments as referenced in [39–41]. The global model consists of three convolutional layers with 64 filters and 3x3 kernels, followed by three fully-connected layers of 256, 128, and the final classification layer. All experiments are run on a cluster of 32 NVIDIA GTX 1080 GPUs.

**Datasets**    To evaluate robust performance in non-i.i.d. data, we set up our experimental environment by distributing the CIFAR-10/100 [21] and SVHN [22] datasets with varying levels of data heterogeneity among clients. We allocated the data to 130 clients based on a Dirichlet distribution with different hyperparameters of $\alpha = [5.0, 0.1]$, following the procedure in pFL-Bench [25]. The lower the $\alpha$, the higher the degree of heterogeneity. Among these clients, only 100 were randomly selected to participate in the training, while the remaining 30 were designated as out-of-distribution (OOD) clients. For the CIFAR10/100, the number of training rounds $R$ was set to 1000. For the FEMNIST, the number of training rounds $R$ was set to 500. For the SVHN, the number of training rounds $R$ was set to 100, 300, and 500, as reported in Table 5. The results reported in the main paper, Table 1, are for the 500 round. In each communication round, only 10 clients are sampled, and the remaining 30 clients serve as out-of-distribution (OOD) clients.

**Hyperparameters**    For the FedAvP algorithm, the hyperparameters include the server policy learning rate $\eta$, client policy learning rate $\lambda$, gradient clipping threshold $c$, and a regularization term $\epsilon$. In our experimentation, we tuned $\eta$ within $[0.4, 0.9]$, $\lambda$ within $[0.1 \sim 0.9]$, $c$ within $[0.4 \sim 1.0]$, and $\epsilon$ within $[0.0 \sim 0.5]$. The validation batch size was also explored within $[64, 128, 192]$. A common hyperparameter across all methods was local epoch set to 5, and local batch is set to 64. The client model learning rate $\gamma$ was searched within the range of $[0.1 \sim 0.3]$. This comprehensive parameter optimization was conducted using an optimization tool known as Optuna[1] [46]. We utilized both the Tree-structured Parzen Estimator algorithm and Random Sampler as hyperparameter samplers within Optuna. The Adaptive Policy Network was consistently implemented in all experiments with FedAvP, comprising the following dense layers: an embedding layer, two hidden layers with 100 neurons each, and an output layer shaped to the $17 \times 17$ distribution size. With FedAvP (Fast Update), it comprised the following dense layers: an embedding layer, two hidden layers with 25 neurons each, and an output layer shaped to the $17 \times 17$ distribution size.

**Baselines**    In all baseline experiments, Random Crop and HorizontalFlip were applied as default augmentations. For the baseline algorithms' hyperparameter settings, specifically for RandAugment (RA) [17], we leveraged a PyTorch implementation [47]. Hyperparameter values were determined in alignment with the procedures described by the authors. For the CIFAR-100 dataset, the hyperparameter $M$ was investigated within the range of $[2, 6, 10, 14]$ and $N$ within $[1, 2]$. For CIFAR-10, the hyperparameter $M$ was investigated within $[4, 5, 7, 9, 11]$ and $N$ within $[2, 3]$. For FEMNIST, the hyperparameter $M$ was investigated within $[4, 5, 7, 9, 11]$ and $N$ within $[1, 2, 3]$. For SVHN, the hyperparameter $M$ was investigated within $[5, 7, 9, 11]$ and $N$ within $[3]$. For TrivialAugment (TA) [24], we utilized the PyTorch library's built-in algorithm [47]. For RandAugment and TrivialAugment, Random Crop and HorizontalFlip were applied first, followed by RandCutout [48]. As for FedMix [8], the hyperparameter $\lambda$ was investigated within $[0.01 \sim 0.1]$. The client model learning rate $\gamma$ was searched within the range of $[0.005 \sim 0.3]$. The hyperparameter $M$ used "All", meaning the average image of all images of each client was used. In the case of FedFA [11], we searched the hyperparameters $p$ and $\alpha$, each within the range of $[0.0 \sim 1.0]$. The client model learning rate $\gamma$ was searched within the range of $[0.005 \sim 0.3]$.

## A.1  Reconstruction Attack Details

We performed experiments using reconstruction attacks [14] to assess if our policy-sharing algorithm offers better privacy in a federated learning context, in comparison to FedMix [8] and FedFA [11], which share data at the input and feature levels respectively. Additionally, we conducted experiments to compare our algorithm with FedGen [9], which shares the label distribution of the client's training data and a generative model that produces a latent from learned latent feature space over the clients' local training data. Despite our algorithm not being specifically designed as a defense against reconstruction attacks, we included ATSPrivacy [27] in our performance evaluation for comparison. ATSPrivacy aims to identify the best policy to counter reconstruction attacks by conducting a policy search. Although ATSPrivacy's primary goal isn't to enhance performance through policy sharing in collaborative learning environments, we chose it as a baseline because it also involves policy search in the context of collaborative learning. We have described below the attack algorithms that utilize input-level, feature-level, and policy gradient information.

---

[1]https://optuna.org/

### A.1.1 Reconstruction Attack using Additional Information

It has been reported that in Collaborative learning systems, Gradient leakage attacks are feasible [14, 42, 43]. These attacks use gradients to reconstruct the training data. We consider a scenario with a central server and clients where learning occurs through the exchange of gradients. Assume we have a given gradient $\nabla W(x, y)$, then we can optimize for a dummy data and label pair $(x', y')$ by minimizing the following objective:

$$x^*, y^* = \arg\min_{x', y'} ||\nabla W(x, y) - \nabla W(x', y')||, \tag{8}$$

where $|| \cdot ||$ denotes a norm distance measure, $\nabla W(x, y)$ is the given gradient, $(x, y)$ is the client's sample data and label, and $(x', y')$ are the targets of optimization. Following [42], we utilize cosine similarity as a cost function instead of norm distance, like in the case of

$$x^*, y^* = \arg\min_{x', y'} \left[ 1 - \ell(\nabla W(x, y), \nabla W(x', y')) \right], \tag{9}$$

where $\ell(x, y) = \langle x, y \rangle / (\|x\| \cdot \|y\|)$.

Here, we assume that input-level information is available, specifically the mean image of the client's data $x_{mean}$ in the FedMix [8]. The method for reconstruction attack utilizing input-level information is as follows:

$$
\begin{aligned}
x^*, y^* = \arg\min_{x', y'} \Big[ &(1 - \alpha_{input}) \cdot (1 - \ell(\nabla W(x, y), \nabla W(x', y'))) \\
&+ \alpha_{input} \cdot \|x' - x_{mean}\| \Big],
\end{aligned}
\tag{10}
$$

where $\alpha_{input}$ is a hyperparameter for the additional term, it is fixed at 0.1 in experiments where input-level information is available and set to 0 in scenarios without input-level information. $x_{mean}$ represents the mean image of the client's data. Cosine similarity was utilized to measure gradient distance, as described in [42]. Similarly, in cases where the client provides feature-level information to the server, as in the FedFA [11], this information can also be used to help with reconstruction. The method for reconstruction attack utilizing feature-level information is as follows:

$$
\begin{aligned}
x^*, y^* = \arg\min_{x', y'} \Big[ &(1 - \alpha_{feat} - \beta_{feat}) \cdot (1 - \ell(\nabla W(x, y), \nabla W(x', y'))) \\
&+ \alpha_{feat} \cdot \mathbb{E}_k \left\| \bar{\mu}^k - \mu^{k'} \right\| \\
&+ \beta_{feat} \cdot \mathbb{E}_k \left\| \bar{\sigma}^k - \sigma^{k'} \right\| \Big],
\end{aligned}
\tag{11}
$$

where $\bar{\mu}^k$ and $\bar{\sigma}^k$ are the momentum updated feature statistics of layer $k$ in the client model, $\mu^{k'}$ and $\sigma^{k'}$ are the feature statistics of layer $k$ with regard to $x'$, $\alpha_{feat}$ and $\beta_{feat}$ are hyperparameters for the additional terms fixed at 0.1 and 0.0, respectively.

In the FedGen algorithm, clients transmit the label distribution of their local training data and the generative model. This information allows us to improve the cost function for reconstruction, which is formulated as follows:

$$
\begin{aligned}
x^*, y^* = \arg\min_{x', y'} \Big[ &(1 - \alpha_{gen} - \beta_{gen}) \cdot (1 - \ell(\nabla W(x, y), \nabla W(x', y'))) \\
&+ \alpha_{gen} \cdot \|c - y'\| \\
&+ \beta_{gen} \cdot (1 - \ell(W^p(z | z \sim G(y')), W(x'))) \Big],
\end{aligned}
\tag{12}
$$

where $G$ is a generative model to generate the latent distribution, $W^p$ refers to the final layer in the client model that takes a latent as an input and outputs prediction logits, $c$ is the label distribution of the client's local training data, $\alpha_{gen}$ and $\beta_{gen}$ are hyperparameters for the additional terms fixed at 0.0001 and 0.0001, respectively. Also, we can initialize $y'$ with $c$.

### A.1.2 Reconstruction Attack using Policy Gradient Information

For the FedAvP algorithm, clients send the policy gradient information. Such policy gradient information can also be utilized as a term of cost function for reconstruction in the following manner:

$$
\begin{aligned}
x^*, y^* = \arg\min_{x', y'} \Big[ &(1 - \alpha_{policy}) \cdot (1 - \ell(\nabla W(x, y), \nabla W(x', y'))) \\
&+ \alpha_{policy} \cdot (1 - \ell(\nabla \ell_{x, y}^{\text{FMPL}}(\theta), \nabla \ell_{x', y'}^{\text{FMPL}}(\theta))) \Big],
\end{aligned}
\tag{13}
$$

where $\alpha_{policy}$ is a hyperparameter for the additional term fixed at 0.1.

When additional information, such as input-level or feature-level, is available in addition to gradient information, it can be utilized, potentially posing a security risk. On the other hand, we have confirmed that even when utilizing the policy gradient information additionally, the reconstruction risk does not increase. The results of the experiments regarding this are summarized in Table 2.

### A.1.3 Experimental details

Our reconstruction attack experiments were conducted on CIFAR-100 with $\alpha = 0.1$, featuring heterogeneous data distribution. Specifically, after 1000 rounds of training, we selected clients for the reconstruction attack. The selection criteria were the client with the most training data (895 data points), Client(L), and the one with the least (156 data points), Client(S). For both clients, we randomly sampled 150 training data points to perform the reconstruction attack. In all experiments, the batch size and reconstruction step for the reconstruction attack were set to 1 and 2400, respectively. We used the Adam optimizer [49]. Gradients obtained from 1 round after 1000 rounds of training were used for the reconstruction attack.

For ATSPrivacy, we trained the 6 policies with FedAvg used in the CIFAR-100 dataset of the author's paper [27], namely 3-1-7, 43-18-18, (3-1-7, 43-18-18), 21-13-3, 7-4-15, and (21-13-3, 7-4-15), in the CIFAR-100 with ($\alpha = 0.1$) environment. We performed the reconstruction attack on the policies (7-4-15) and (21-13-3, 7-4-15), which showed the best accuracy performance. The (21-13-3, 7-4-15) policy is a hybrid policy described in the author's paper [27], and we applied it by randomly sampling one of the two policies.

## A.2  Additional Results with a Larger Model

We conducted experiments on a model with more parameters than those used in the main experiments. The model used in these experiments is a simplified version of VGG11 [50, 51], where all dropout and batch normalization layers are removed, and the filters and the size of all fully-connected layers are reduced by a factor of 2. This model contains about three times more network parameters than the networks used in the main paper. We verify the performance of our model on non-IID datasets, specifically SVHN-10 ($\alpha = 0.1$), with different communication rounds.

| Method | $R = 100$ | | $R = 300$ | | $R = 500$ | |
|---|---|---|---|---|---|---|
| | Test (%) | OOD (%) | Test (%) | OOD (%) | Test (%) | OOD (%) |
| FedAvg+Default | 76.01 | 74.18 | 83.92 | 82.59 | 90.38 | 91.64 |
| FedAvg+RandAugment | 59.30 | 53.52 | 81.04 | 77.74 | 89.75 | 89.96 |
| FedAvg+TrivialAugment | 44.74 | 40.79 | 78.83 | 75.59 | 89.51 | 89.80 |
| FedExP+Default | 84.89 | 84.97 | 87.17 | 87.44 | 90.03 | 90.72 |
| FedExP+RandAugment | 74.44 | 71.87 | 87.56 | 85.31 | 88.20 | 88.92 |
| FedExP+TrivialAugment | 43.56 | 40.47 | 83.26 | 81.51 | 88.07 | 88.68 |
| FedFA+Default | 83.19 | 83.10 | 88.99 | 89.45 | 91.18 | 92.03 |
| FedFA+RandAugment | 62.62 | 63.74 | 86.77 | 86.23 | 90.92 | 91.97 |
| FedFA+TrivialAugment | 8.477 | 10.63 | 68.42 | 70.29 | 86.87 | 88.09 |
| FedAvP (Fast Update) | **86.14** | **87.24** | **91.56** | **92.13** | **93.85** | **93.34** |

Table 5: Classification accuracies with the different communication rounds of $R = [100, 300, 500]$ in SVHN dataset with $\alpha = 0.1$ using the VGG11s model.

## A.3 Fast Update Algorithm

To reduce computation for local clients from joint learning, we perform periodic policy updates, specifically when $n \mod \tau == 0$. In all our experiments, we set $\tau = 5$ and reduced the hidden size of the policy neural network to two 25-dimensional hidden layers with FedAvP (Fast Update) algorithm.

---

**Algorithm 2** FedAvP (Fast Update) : Joint Training

---

**Input:** # of communication round $R$, # of client $N$, server policy learning rate $\eta$, client model learning rate $\gamma$, client policy learning rate $\lambda$, local steps $E$, gradient clipping threshold $c$, regularization term $\epsilon$.

Initialize the global model parameter $w_{g_r}$ and the global policy parameter $\theta_{g_r}$

**for** $r = 1, ..., R$ **do**
    Sample $K$ clients from $1, ..., N$ clients
    **for** $k = 1, ..., K$ **do**
        Set $w_0^k = w_{g_r}$ and $\theta_0^k = \theta_{g_r}$
        $w_*^k, \theta_*^k \leftarrow$ LOCALUPDATE$(w_0^k, \theta_0^k)$
    $w_{g_{r+1}} \leftarrow \Sigma_k \alpha_k w_*^k$
    $\Delta\theta \leftarrow \Sigma_k \alpha_k (\theta_*^k - \theta_{g_r})$
    $\theta_{g_{r+1}} \leftarrow \theta_{g_r} + \eta \Delta\theta$

**procedure** LOCALUPDATE$(w, \theta)$
    Initialize $w_0^k = w, \theta_0^k = \theta$
    **for** Each local step $n$ from 0 to $E$ **do**
        Sample transformations $t_{p_{\theta_n}}$ from $p_{\theta_n}$
        Sample batch data $t_{p_{\theta_n}}(D_{k,n}^{\text{train}})$ from $D_k$ and $t_{p_{\theta_n}}$
        Compute $\nabla\ell_{t_{p_{\theta_n}}(D_{k,n}^{\text{train}})}(w_n^k)$ using Eq.(3)
        $w_{n+1}^k \leftarrow w_n^k - \gamma\nabla\ell_{t_{p_{\theta_n}}(D_{k,n}^{\text{train}})}(w_n^k)$
        **if** $n \mod \tau == 0$ **then**
            Sample batch data $D_{k,n}^{\text{val}}$ from $D_k$
            Compute $\nabla\ell_{D_{k,n}^{\text{val}}}(w_{n+1}^k)$ at $w_{n+1}^k$
            $\theta_{n+1}^k \leftarrow \theta_n^k - \lambda\nabla\ell_{D_{k,n}^{\text{val}}}^{\text{FMPL}}(\theta_n^k)$ using Prop.1
    Set $w_*^k = w_{n+1}^k$ and $\theta_*^k = \theta_{n+1}^k$
    Send $w_*^k$ and $\theta_*^k$ to the server

---

## A.4 Additional Results of Computation cost

For additional computational comparisons, we measured the time taken to reach a target accuracy of 40% on the CIFAR-100 dataset with $\alpha = 5.0$ and the time taken to reach a target accuracy of 75% on the CIFAR-10 dataset with $\alpha = 5.0$. In the case of CIFAR-100 with $\alpha = 5.0$, FedAvP (Fast Update) achieved results more than 1 hour faster than FedAvg. Our method generally produced faster results compared to applying RandAugment and TrivialAugment which enhanced the performance of FedAvg in table 1.

| Method | CIFAR-100 ($\alpha = 5.0$) | |
| --- | --- | --- |
| | Rounds (40%) | Time (40%) |
| FedAvg + Default | 600 | 2.90 hours |
| FedAvg + RA | 300 | 1.90 hours |
| FedAvg + TA | 400 | 2.10 hours |
| FedAvP (Fast Update) | 200 | 1.50 hours |
| FedAvP | 150 | 3.39 hours |

Table 6: Computation Time in CIFAR-100

| Method | CIFAR-10 ($\alpha = 5.0$) | |
| --- | --- | --- |
| | Rounds (75%) | Time (75%) |
| FedAvg + Default | 200 | 0.77 hours |
| FedAvg + RA | 200 | 1.35 hours |
| FedAvg + TA | 350 | 1.68 hours |
| FedAvP (Fast Update) | 150 | 1.08 hours |
| FedAvP | 150 | 2.00 hours |

Table 7: Computation Time in CIFAR-10

## A.5 Ablation Results of Hyperparameters $\epsilon$

| | $\epsilon = 0.1$ | $\epsilon = 0.3$ | $\epsilon = 0.4$ | $\epsilon = 0.5$ |
| --- | --- | --- | --- | --- |
| Method | Test (%) | Test (%) | Test (%) | Test (%) |
| FedAvP | 83.36 | 83.38 | 83.98 | 83.09 |

Table 8: CIFAR-10 with $\alpha = 5.0$.

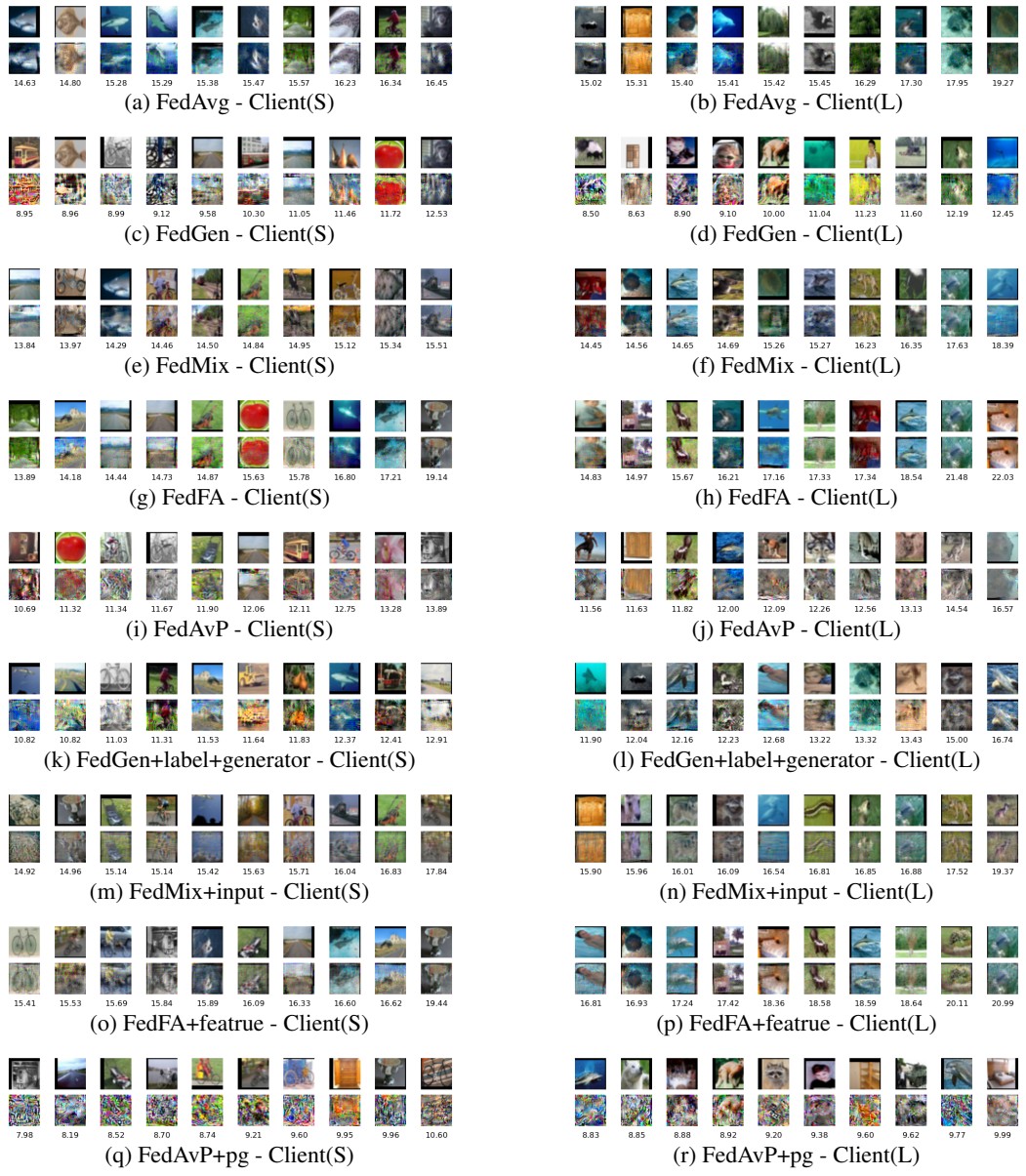

Figure 4: Results of the reconstruction attacks in Table 2. The first row of each result represents the random client's training samples, and the second row is the reconstructed samples by the server. We visualized high-PSNR samples selected from random samples. The numbers below indicate the PSNR values of the reconstructed samples.

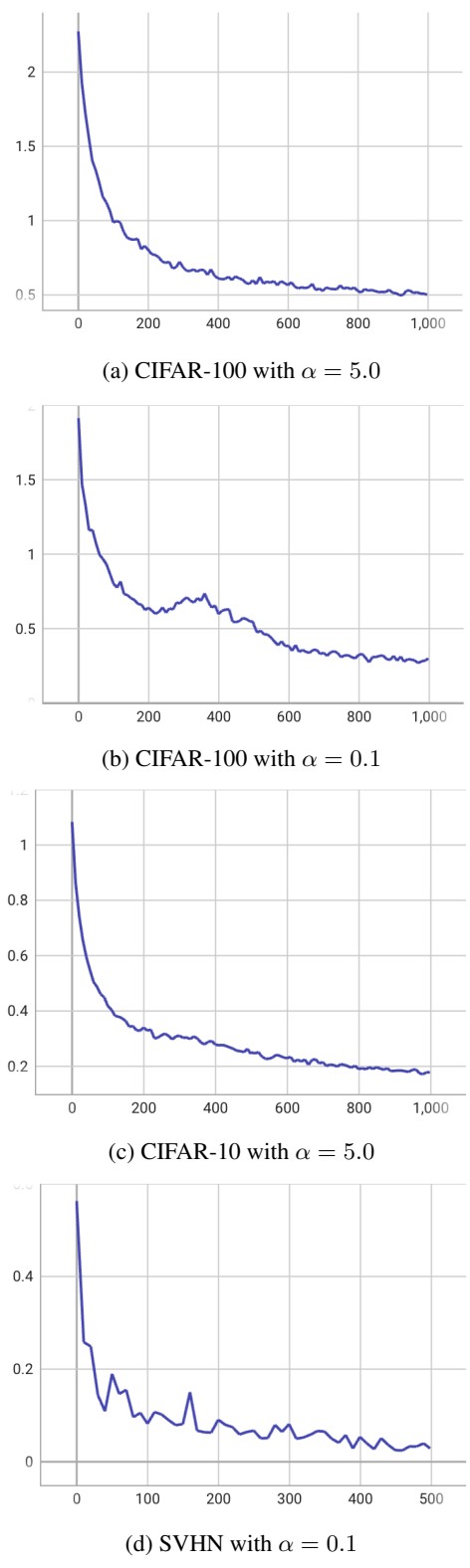

(a) CIFAR-100 with $\alpha = 5.0$

(b) CIFAR-100 with $\alpha = 0.1$

(c) CIFAR-10 with $\alpha = 5.0$

(d) SVHN with $\alpha = 0.1$

Figure 5: Training loss convergence of our FedAvP algorithm

## A.6 Additional Results of Non-i.i.d. Experiments

| | Dataset | CIFAR-100 | | CIFAR-10 | | SVHN | | FEMNIST |
|---|---|---|---|---|---|---|---|---|
| | | $\alpha = 5.0$ | $\alpha = 0.1$ | $\alpha = 5.0$ | $\alpha = 0.1$ | $\alpha = 5.0$ | $\alpha = 0.1$ | |
| | Method | Test (%) | Test (%) | Test (%) | Test (%) | Test (%) | Test (%) | Test (%) |
| FedAvg | + Default | 40.05 | 37.34 | 79.76 | 72.60 | 92.78 | 85.58 | 80.65 |
| | + RandAugment | 47.29 | 43.60 | 82.82 | 73.73 | 92.48 | 84.84 | 79.40 |
| | + TrivialAugment | 46.61 | 42.16 | 82.00 | 71.09 | 91.99 | 83.36 | 79.01 |
| FedProx | + Default | 40.57 | 37.71 | 80.64 | 73.23 | 93.15 | 86.79 | 81.45 |
| | + RandAugment | 45.97 | 41.39 | 82.56 | 73.71 | 92.33 | 85.52 | 77.11 |
| | + TrivialAugment | 46.61 | 41.81 | 81.83 | 70.89 | 91.67 | 84.11 | 79.67 |
| FedDyn | + Default | 42.09 | 38.52 | 80.36 | 73.86 | 93.16 | 87.60 | 80.47 |
| | + RandAugment | 45.70 | 42.24 | 82.51 | 72.78 | 92.16 | 81.47 | 77.64 |
| | + TrivialAugment | 46.83 | 41.10 | 82.03 | 70.34 | 92.22 | 83.41 | 79.31 |
| FedExP | + Default | 42.76 | 38.28 | 80.64 | 73.70 | 92.77 | 86.66 | 81.45 |
| | + RandAugment | 46.13 | 42.23 | 82.86 | 70.78 | 92.12 | 84.63 | 79.69 |
| | + TrivialAugment | 48.55 | 42.09 | 82.51 | 71.07 | 92.64 | 83.72 | 80.20 |
| FedGen | + Default | 42.14 | 38.27 | 80.23 | 72.74 | 92.71 | 86.79 | 81.86 |
| | + RandAugment | 47.11 | 43.10 | 81.90 | 73.42 | 91.84 | 84.39 | 79.34 |
| | + TrivialAugment | 47.71 | 40.76 | 82.58 | 70.87 | 91.73 | 83.23 | 77.35 |
| FedMix | + Default | 40.26 | 38.69 | 80.99 | 74.54 | 92.80 | 86.02 | 81.63 |
| | + RandAugment | 46.69 | 43.00 | 83.08 | 74.25 | 92.36 | 83.44 | 79.46 |
| | + TrivialAugment | 46.64 | 42.63 | 81.83 | 71.50 | 91.85 | 82.34 | 77.84 |
| FedFA | + Default | 43.70 | 41.21 | 82.61 | 76.02 | 92.77 | 87.33 | 81.13 |
| | + RandAugment | 48.86 | 43.44 | 82.44 | 73.53 | 91.21 | 81.32 | 78.71 |
| | + TrivialAugment | 47.86 | 43.45 | 80.12 | 72.89 | 91.89 | 78.62 | 78.96 |
| FedAvP | (W/ Local Policy) | 49.04 | 43.86 | 83.64 | 73.43 | 94.71 | 87.05 | 83.94 |
| FedAvP | (Fast Update) | 49.97 | 45.08 | 83.55 | **77.20** | **95.14** | 87.86 | **84.47** |
| FedAvP | | **50.47** | **45.96** | **83.78** | 77.10 | 95.02 | **89.81** | 84.27 |

Table 9: Classification accuracies with different heterogeneity degrees ($\alpha = 5.0$ and $\alpha = 0.1$) across CIFAR-100/10, SVHN, and FEMNIST datasets.

## A.7 The Scalability of FedAvP

| Method | Round 100 | Round 300 | Round 500 |
|---|---|---|---|
| FedAvP (Fast Update) / 2 layers | **86.85** | 87.84 | 87.86 |
| FedAvP (Fast Update) / 3 layers | 84.04 | **89.72** | **92.07** |

Table 10: Test accuracy (%) of FedAvP (Fast Update) on SVHN with $\alpha = 0.1$ across different rounds and search spaces.

| Method | Round 100 | Round 300 | Round 500 |
|---|---|---|---|
| FedAvP (Fast Update) / 2 layers | **92.76** | **94.67** | **95.14** |
| FedAvP (Fast Update) / 3 layers | 92.73 | 94.44 | 95.01 |

Table 11: Test accuracy (%) of FedAvP (Fast Update) on SVHN with $\alpha = 5.0$ across different rounds and search spaces.

## A.8 Additional Results with ViT-T

We present additional results using the ViT-T model[52, 53] on CIFAR-100 with different heterogeneity degrees.

| Method | CIFAR-100 / 5.0 | CIFAR-100 / 0.1 |
|---|---|---|
| FedAvg + Default | 31.75 | 30.71 |
| FedAvg + RandAugment | 42.39 | 41.70 |
| FedAvg + TrivialAugment | 41.58 | 33.75 |
| FedExp + Default | 37.33 | 35.77 |
| FedExp + RandAugment | 46.36 | 45.50 |
| FedExp + TrivialAugment | 44.37 | 40.08 |
| FedAvP (Fast Update) | **51.10** | **47.85** |

Table 12: Test accuracy (%) using ViT-T on CIFAR-100 with $\alpha = 5.0$ and $\alpha = 0.1$.

### A.8.1 Computation Time of ViT-T on CIFAR-100

| Method | Rounds (30%) | Time (30%) |
|---|---|---|
| FedAvg + Default | 400 | 3.08 hours |
| FedAvg + RandAugment | 300 | 2.18 hours |
| FedAvg + TrivialAugment | 400 | 2.63 hours |
| FedAvP (Fast Update) | 220 | 3.23 hours |

Table 13: Computation time and rounds to reach 30% test accuracy on CIFAR-100 with $\alpha = 5.0$.

| Method | Rounds (25%) | Time (25%) |
|---|---|---|
| FedAvg + Default | 300 | 4.33 hours |
| FedAvg + RandAugment | 400 | 6.02 hours |
| FedAvg + TrivialAugment | 480 | 8.27 hours |
| FedAvP (Fast Update) | 260 | 3.73 hours |

Table 14: Computation time and rounds to reach 25% test accuracy on CIFAR-100 with $\alpha = 0.1$.

## A.9 Comparison with Other Classic Non-IID Methods

We conducted additional experiments with non-IID algorithms, including FedNova [54] and SCAFFOLD [55].

| Method | CIFAR-100 / 0.1 | CIFAR-10 / 0.1 | SVHN / 0.1 | FEMNIST |
|---|---|---|---|---|
| FedNova + Default | 38.52 | 74.45 | 88.16 | 81.21 |
| FedNova + RandAugment | 42.43 | 74.08 | 84.42 | 79.79 |
| FedNova + TrivialAugment | 40.23 | 71.99 | 82.96 | 78.92 |
| SCAFFOLD + Default | 44.94 | 75.67 | 87.26 | 83.17 |
| SCAFFOLD + RandAugment | 43.57 | 72.40 | 77.07 | 79.31 |
| SCAFFOLD + TrivialAugment | 42.14 | 64.12 | 14.70 | 78.06 |
| FedAvP (Fast Update) | 45.08 | **77.20** | 87.86 | **84.47** |
| FedAvP | **45.96** | 77.10 | **89.81** | 84.27 |

Table 15: Test accuracy (%) on various datasets under non-IID settings with $\alpha = 0.1$.

## A.10 Additional Results of Non-i.i.d. Experiments with Equally-Weighted Metric

In the main paper, we followed the weighted accuracy metric as described in the pFL-Bench [25]. Here, we provide additional results for non-i.i.d. experiments using an equally-weighted metric on CIFAR-100.

|  | Method | CIFAR-100 / 5.0 | CIFAR-100 / 0.1 |
|---|---|---|---|
| FedAvg | + Default | 40.04 | 36.98 |
|  | + RandAugment | 47.30 | 43.17 |
|  | + TrivialAugment | 46.61 | 42.04 |
| FedProx | + Default | 40.56 | 37.61 |
|  | + RandAugment | 45.95 | 41.25 |
|  | + TrivialAugment | 46.59 | 41.67 |
| FedDyn | + Default | 42.11 | 38.23 |
|  | + RandAugment | 45.68 | 42.08 |
|  | + TrivialAugment | 46.84 | 40.92 |
| FedExp | + Default | 42.78 | 38.22 |
|  | + RandAugment | 46.14 | 41.97 |
|  | + TrivialAugment | 48.54 | 42.01 |
| FedGen | + Default | 42.12 | 38.05 |
|  | + RandAugment | 47.11 | 42.96 |
|  | + TrivialAugment | 47.73 | 40.62 |
| FedMix | + Default | 39.59 | 38.46 |
|  | + RandAugment | 46.67 | 42.70 |
|  | + TrivialAugment | 46.62 | 42.49 |
| FedFA | + Default | 43.68 | 41.18 |
|  | + RandAugment | 48.87 | 43.26 |
|  | + TrivialAugment | 47.86 | 43.36 |
| FedAvP | (W/ Local Policy) | 49.05 | 43.64 |
| FedAvP | (Fast Update) | 49.94 | 45.09 |
| FedAvP |  | **50.59** | **45.93** |

Table 16: Test accuracy (%) on CIFAR-100 with $\alpha = 5.0$ and $\alpha = 0.1$ using an equally-weighted metric.

## A.11 Experiments on Extreme label-skew Settings

The table below presents the results across different datasets and partitioning strategies, specifically the quantity-based label skew settings described in [56]. Here, $C$ is the number of different labels held by each client. In extreme label skew cases, such as $C = 1$, where data labels are highly partitioned, our algorithm shows slightly lower performance on CIFAR-100 ($C = 1$). However, in all other cases, our algorithm demonstrates improved performance.

| Method | CIFAR-100 | | | SVHN | | |
|---|---|---|---|---|---|---|
| | $C = 3$ | $C = 2$ | $C = 1$ | $C = 3$ | $C = 2$ | $C = 1$ |
| FedAvg + Default | 27.75 | 24.55 | **7.59** | 89.50 | 85.34 | 8.45 |
| FedAvg + RandAugment | 25.38 | 22.94 | 6.69 | 85.75 | 79.05 | 7.64 |
| FedAvg + TrivialAugment | 24.36 | 19.58 | 4.84 | 85.35 | 77.99 | 7.64 |
| FedProx + Default | 27.10 | 24.26 | 7.51 | 89.18 | 85.87 | 9.39 |
| FedProx + RandAugment | 26.10 | 24.46 | 5.66 | 86.19 | 80.11 | 7.63 |
| FedProx + TrivialAugment | 24.15 | 20.14 | 3.17 | 84.73 | 78.56 | 8.34 |
| FedDyn + Default | 27.84 | 24.89 | 7.39 | 89.59 | 86.65 | 14.56 |
| FedDyn + RandAugment | 25.80 | 23.34 | 1.57 | 83.64 | 80.06 | 9.52 |
| FedDyn + TrivialAugment | 24.50 | 19.70 | 3.81 | 84.34 | 79.06 | 9.40 |
| FedFA + Default | 27.51 | 23.23 | 6.83 | 89.91 | 82.94 | 11.60 |
| FedFA + RandAugment | 21.58 | 25.13 | 3.09 | 87.97 | 59.52 | 11.60 |
| FedFA + TrivialAugment | 23.33 | 20.07 | 5.58 | 87.05 | 68.69 | 11.87 |
| SCAFFOLD + Default | 29.75 | 20.09 | 1.18 | 90.07 | 85.02 | 9.39 |
| SCAFFOLD + RandAugment | 26.56 | 17.06 | 1.18 | 82.57 | 6.52 | 14.45 |
| SCAFFOLD + TrivialAugment | 19.21 | 12.02 | 1.17 | 79.17 | 6.53 | 7.64 |
| FedAvP (Local) | 27.74 | 24.35 | 5.38 | 91.74 | 88.74 | 11.60 |
| FedAvP (Fast Update) | **31.54** | **30.96** | 6.17 | **92.53** | **90.13** | **18.92** |

Table 17: Test accuracy (%) on CIFAR-100 and SVHN datasets under quantity-based label skew settings. $C$ denotes the number of different labels held by each client.

# B Proof of Proposition 1

Consider the federated meta-policy loss derived from the updated weight $w_n^k$ for client $k$ at step $n$ using a first-order Taylor expansion:

$$\ell_{D_k^{\text{val}}}(w_{g_{r+1}}) \approx \ell_{D_k^{\text{val}}}(w_n^k) + \nabla\ell_{D_k^{\text{val}}}(w_n^k)^T(w_{g_{r+1}} - w_n^k) \tag{14}$$

When calculating the policy gradient of this loss with respect to $\theta_{n-1}^k$ for client $k$, the first-order gradient approximation is as follows:

$$-\alpha_k \cdot lr \frac{\partial(\nabla\ell_{D_k^{\text{val}}}(w_n^k)^T \nabla\ell_{t_{P_{\theta_{n-1}^k}}(D_{k,n-1}^{\text{train}})}(w_{n-1}^k))}{\partial\theta_{n-1}^k}, \tag{15}$$

$$\text{where } w_n^k = w_{g_r} - lr \cdot g_{w_0^k}^{\text{aug}} - \ldots - lr \cdot g_{w_{n-1}^k}^{\text{aug}}, \tag{16}$$

$$g_{w_n^k}^{\text{aug}} = \nabla\ell_{t_{P_{\theta_k}}(D_{k,n}^{\text{train}})}(w_n^k), \tag{17}$$

$$w_{g_{r+1}} = \sum \alpha_k w_N^k. \tag{18}$$

*Proof.*

$$\frac{\partial\ell_{D_k^{\text{val}}}(w_{g_{r+1}})}{\partial\theta_{n-1}^k} \approx \frac{\partial\ell_{D_k^{\text{val}}}(w_n^k)}{\partial\theta_{n-1}^k} + \frac{\partial(\nabla\ell_{D_k^{\text{val}}}(w_n^k)^T(w_{g_{r+1}} - w_n^k))}{\partial\theta_{n-1}^k} \quad \text{(Taylor's theorem)} \tag{19}$$

First, let's calculate starting from the left term in Eq. (19)

$$\frac{\partial\ell_{D_k^{\text{val}}}(w_n^k)}{\partial\theta_{n-1}^k} \tag{20}$$

$$= \nabla\ell_{D_k^{\text{val}}}(w_n^k)^T \cdot \frac{\partial w_n^k}{\partial\theta_{n-1}^k} \quad \text{(using the chain rule)} \tag{21}$$

$$= \nabla\ell_{D_k^{\text{val}}}(w_n^k)^T \cdot \frac{\partial(w_{n-1}^k - lr \cdot g_{w_{n-1}^k}^{\text{aug}})}{\partial\theta_{n-1}^k} \quad \text{(using the definition of } w_n^k \text{ in (16))} \tag{22}$$

$$= -lr \cdot \nabla\ell_{D_k^{\text{val}}}(w_n^k)^T \frac{\partial(g_{w_{n-1}^k}^{\text{aug}})}{\partial\theta_{n-1}^k} \quad \left(\text{using } \frac{\partial w_{n-1}^k}{\partial\theta_{n-1}^k} = 0\right) \tag{23}$$

$$= -lr \cdot \frac{\partial(\nabla\ell_{D_k^{\text{val}}}(w_n^k)^T \cdot \nabla\ell_{t_{P_{\theta_{n-1}^k}}(D_{k,n-1}^{\text{train}})}(w_{n-1}^k))}{\partial\theta_{n-1}^k} \quad \text{(using the definition of } g_{w_{n-1}^k}^{\text{aug}} \text{ in (17) )} \tag{24}$$

Next, we calculate the right term in Eq. (19)

$$\frac{\partial(\nabla\ell_{D_k^{\mathrm{val}}}(w_n^k)^T(w_{g_{r+1}} - w_n^k))}{\partial\theta_{n-1}^k}$$

$$= \frac{\partial w_n^k}{\partial\theta_{n-1}^k} \cdot \frac{\partial\nabla\ell_{D_k^{\mathrm{val}}}(w_n^k)^T}{\partial w_n^k} \cdot (w_{g_{r+1}} - w_n^k) + \nabla\ell_{D_k^{\mathrm{val}}}(w_n^k)^T \cdot \frac{\partial(w_{g_{r+1}} - w_n^k)}{\partial\theta_{n-1}^k}$$

(using the chain rule and $(fg)' = f'g + fg'$)

$$\tag{25}$$

$$= \nabla\ell_{D_k^{\mathrm{val}}}(w_n^k)^T \cdot \frac{\partial(w_{g_{r+1}} - w_n^k)}{\partial\theta_{n-1}^k}$$

(treating $\nabla\ell_{D_k^{\mathrm{val}}}(w_n^k)$ as a constant for first-order)

$$\tag{26}$$

$$= \nabla\ell_{D_k^{\mathrm{val}}}(w_n^k)^T \cdot \frac{\partial(\alpha_k \cdot w_N^k - w_n^k)}{\partial\theta_{n-1}^k}$$

(using the definition of $w_{g_{r+1}}$ in (18))

$$\tag{27}$$

$$= \nabla\ell_{D_k^{\mathrm{val}}}(w_n^k)^T \cdot \frac{\partial((\alpha_k - 1) \cdot w_n^k)}{\partial\theta_{n-1}^k}$$

(using the definition of $w_N^k$ in (16)) and treating $g_{w_n^k}^{\mathrm{aug}}$ as a constant for first-order)

$$\tag{28}$$

$$= (\alpha_k - 1) \cdot \nabla\ell_{D_k^{\mathrm{val}}}(w_n^k)^T \cdot \frac{\partial(w_n^k)}{\partial\theta_{n-1}^k} \tag{29}$$

$$= (\alpha_k - 1) \cdot \nabla\ell_{D_k^{\mathrm{val}}}(w_n^k)^T \cdot \frac{\partial(w_{n-1}^k - lr \cdot g_{w_{n-1}^k}^{\mathrm{aug}})}{\partial\theta_{n-1}^k} \tag{30}$$

(using the definition of $w_n^k$ in (16))

$$= (\alpha_k - 1) \cdot \nabla\ell_{D_k^{\mathrm{val}}}(w_n^k)^T \cdot \frac{\partial(-lr \cdot g_{w_{n-1}^k}^{\mathrm{aug}})}{\partial\theta_{n-1}^k}$$

(using $\frac{\partial w_{n-1}^k}{\partial\theta_{n-1}^k} = 0$ )

$$\tag{31}$$

$$= -lr \cdot (\alpha_k - 1) \cdot \nabla\ell_{D_k^{\mathrm{val}}}(w_n^k)^T \cdot \frac{\partial(g_{w_{n-1}^k}^{\mathrm{aug}})}{\partial\theta_{n-1}^k} \tag{32}$$

$$= -lr \cdot (\alpha_k - 1) \cdot \frac{\partial(\nabla\ell_{D_k^{\mathrm{val}}}(w_n^k)^T \cdot \nabla\ell_{t_{p_{\theta_{n-1}^k}}(D_{k,n-1}^{\mathrm{train}})}(w_{n-1}^k))}{\partial\theta_{n-1}^k}$$

(using the definition of $g_{w_{n-1}^k}^{\mathrm{aug}}$ in (17) )

$$\tag{33}$$

By combining Eq. (24) and Eq. (33),

$$\frac{\partial\ell_{D_k^{\mathrm{val}}}(w_{g_{r+1}})}{\partial\theta_{n-1}^k} \approx -\alpha_k \cdot lr \frac{\partial(\nabla\ell_{D_k^{\mathrm{val}}}(w_n^k)^T\nabla\ell_{t_{p_{\theta_{n-1}^k}}(D_{k,n-1}^{\mathrm{train}})}(w_{n-1}^k))}{\partial\theta_{n-1}^k} \tag{34}$$

$$\square$$

# C Proof of Adaptive Policy Search

$$\theta_{g_{r+1}} \approx \theta_{g_r} - \eta\lambda\frac{\partial}{\partial\theta_0^k}\mathbb{E}\left[\sum_{j=0}^{n}L_{k,j} - \frac{\lambda}{2}\sum_{j=0}^{n}\sum_{s=0}^{j-1}\langle\nabla L_{k,j}\cdot\nabla L_{k,s}\rangle\right], \tag{35}$$

where $L_{k,j} \equiv \ell_{D_{k,j}^{\text{val}}}^{\text{FMPL}}(\theta_0^k)$ represents the the federated meta-policy loss in computed on the client $k$'s $j$-th validation data batch using the global policy parameters $\theta_0^k$ received from the server. $\langle\nabla L_{k,j}\cdot\nabla L_{k,s}\rangle$ represents the dot-product between policy gradients on the client $k$.

*Proof.* We refer to the proof of reptile [30, 57] and use the following definitions:

$$\theta_{g_{r+1}} = \theta_{g_r} + \eta\sum\alpha_k(\theta_*^k - \theta_{g_r}) \qquad \text{(server policy update rule)} \tag{36}$$

$$\theta_{n+1}^k = \theta_n^k - \lambda\ell_{F_n}'(\theta_n^k) \qquad \text{(client policy update rule)} \tag{37}$$

$$g_n^k = \ell_{F_n}'(\theta_n^k) \qquad \text{(gradient during SGD using } \ell_{F_n} = \ell_{D_{k,n}^{\text{val}}}^{\text{FMPL}}) \tag{38}$$

$$\bar{g}_n^k = \ell_{F_n}'(\theta_0^k) \qquad \text{(gradient at initial policy } \theta_0^k = \theta_{g_r}) \tag{39}$$

$$\overline{H}_n^k = \ell_{F_n}''(\theta_0^k) \qquad \text{(hessian at initial policy } \theta_0^k = \theta_{g_r}) \tag{40}$$

$$\theta_{g_r} = \theta_0^k \qquad \text{(initial policy at client)} \tag{41}$$

$$\theta_*^k = \theta_{n+1}^k \qquad \text{(updated policy at client)} \tag{42}$$

$$L_{k,j} = \ell_{D_{k,j}^{\text{val}}}^{\text{FMPL}}(\theta_0^k) = \ell_{F_j} \qquad \text{(abbreviation for FMPL)} \tag{43}$$

First, let's calculate $g_n^k$ in Eq. (38) to $O(\lambda^2)$ as follows:

$$g_n^k = \ell_{F_n}'(\theta_0^k) + \ell_{F_n}''(\theta_0^k)(\theta_n^k - \theta_0^k) + O(\lambda^2) \qquad \text{(Taylor's theorem)} \tag{44}$$

$$= \bar{g}_n^k + \overline{H}_n^k(\theta_n^k - \theta_0^k) + O(\lambda^2) \qquad \text{(using the definitions of (39) and (40))} \tag{45}$$

$$= \bar{g}_n^k - \lambda\overline{H}_n^k\sum_{j=0}^{n-1}g_j^k + O(\lambda^2) \qquad \text{(using } \theta_n^k - \theta_0^k = -\lambda\sum_{j=0}^{n-1}g_j^k) \tag{46}$$

$$= \bar{g}_n^k - \lambda\overline{H}_n^k\sum_{j=0}^{n-1}\bar{g}_j^k + O(\lambda^2) \qquad \text{(using } g_j^k = \bar{g}_j^k + O(\lambda^2)) \tag{47}$$

$$\tag{48}$$

Then, we define the following in (36):

$$\theta_*^k = \theta_{g_r} - \lambda g_{inner}^k \tag{49}$$

$$\theta_{n+1}^k = \theta_0^k - \lambda g_{inner}^k \qquad \text{(using the definitions of (41) and (42) )} \tag{50}$$

$$g_{inner}^k = \sum_{j=0}^{n}\ell_{F_j}'(\theta_j^k) \qquad \text{(using } \theta_{n+1}^k = \theta_0^k - \lambda\sum_{j=0}^{n}\ell_{F_j}'(\theta_j^k)) \tag{51}$$

$$= \sum_{j=0}^{n}g_j^k \qquad \text{(using the definition of } \ell_{F_j}'(\theta_j^k) \text{ in (38))} \tag{52}$$

$$\approx \sum_{j=0}^{n}\ell_{F_j}'(\theta_0^k) - \lambda\sum_{j=0}^{n}\sum_{s=0}^{j-1}\ell_{F_j}''(\theta_0^k)\cdot\ell_{F_s}'(\theta_0^k) \qquad \text{(using the definition of } g_j^k \text{ in (47))} \tag{53}$$

By combining Eq. (36) and Eq. (49) and taking the expectation over the clients and local updates,

$$\theta_{g_{r+1}} \approx \theta_{g_r} + \eta \mathbb{E}[-\lambda g_{inner}^k] \tag{54}$$

$$\approx \theta_{g_r} - \eta \lambda \mathbb{E}[\sum_{j=0}^{n} \ell'_{F_j}(\theta_0^k) - \lambda \sum_{j=0}^{n} \sum_{s=0}^{j-1} \ell''_{F_j}(\theta_0^k) \cdot \ell'_{F_s}(\theta_0^k)] \qquad \text{(using Eq. (53))}$$

$$\tag{55}$$

$$\approx \theta_{g_r} - \eta \lambda \mathbb{E}[\sum_{j=0}^{n} \ell'_{F_j}(\theta_0^k) - \frac{\lambda}{2} \sum_{j=0}^{n} \sum_{s=0}^{j-1} (\ell''_{F_j}(\theta_0^k) \cdot \ell'_{F_s}(\theta_0^k) + \ell'_{F_j}(\theta_0^k) \cdot \ell''_{F_s}(\theta_0^k))] \tag{56}$$

$$\approx \theta_{g_r} - \eta \lambda \frac{\partial}{\partial \theta_0^k} \mathbb{E}\left[ \sum_{j=0}^{n} L_{k,j} - \frac{\lambda}{2} \sum_{j=0}^{n} \sum_{s=0}^{j-1} \langle \nabla L_{k,j} \cdot \nabla L_{k,s} \rangle \right] \qquad \text{(using Eq. (43))}$$

$$\tag{57}$$

$$\square$$

