# OpenReview forum: "FedAvP: Augment Local Data via Shared Policy in Federated Learning"
_NeurIPS.cc/2024/Conference — NeurIPS 2024 poster_

### Official Review · Reviewer_866Z · 2024-06-13

**Soundness:** 3
**Presentation:** 3
**Contribution:** 3
**Rating:** 6
**Confidence:** 2

**Summary:**

This paper proposes FedAvP, a novel algorithm to perform data augmentation via shared policy in federated learning (FL). Extensive experiments verify the effectiveness of the proposed algorithm.

**Strengths:**

S1: The proposed algorithm is novel with solid theoretical analysis.

S2: The experiments are comprehensive, with open-source code submitted.

**Weaknesses:**

W1: The authors can conduct more experiments on more comprehensive non-IID settings. For example, in a previous FL benchmark on non-IID data [1], the quantity-based label skew settings are challenging. The authors are suggested to experiment on such settings to further support the effectiveness of the proposed algorithm.

[1] Li, Qinbin, et al. "Federated learning on non-iid data silos: An experimental study." 2022 IEEE 38th international conference on data engineering (ICDE). IEEE, 2022.

**Questions:**

Please see weaknesses. The authors are suggested to conduct more comprehensive experiments. I will adjust the score based on the author response.

**Limitations:**

Please see weaknesses.

---

> ### Author Rebuttal · Authors · 2024-08-07
>
> **W1. More experiments on FL benchmark on non-IID data**
>
> We conduct experiments on comprehensive non-IID settings suggested by the reviewer, specifically the quantity-based label skew settings described in [1]. The table below presents the results across different datasets and partitioning strategies. Here, C is the number of different labels held by each client. In extreme label skew cases, such as  C = 1, where data labels are highly partitioned, our algorithm shows slightly lower performance on CIFAR-100 (C = 1). However, in all other cases, our algorithm demonstrates improved performance.
>
> |Niid-bench(dataset/C)[1]|CIFAR100(C=3)|CIFAR100(C=2)|CIFAR100(C=1)|SVHN(C=3)|SVHN(C=2)|SVHN(C=1)|
> |-|-|-|-|-|-|-|
> |FedAvg+Default|27.75|24.55|**7.59**|89.5|85.34|8.45|
> |FedAvg+RandAugment|25.38|22.94|6.69|85.75|79.05|7.64|
> |FedAvg+TrivialAugment|24.36|19.58|4.84|85.35|77.99|7.64|
> |FedProx+Default|27.1|24.26|7.51|89.18|85.87|9.39|
> |FedProx+RandAugment|26.1|24.46|5.66|86.19|80.11|7.63|
> |FedProx+TrivialAugment|24.15|20.14|3.17|84.73|78.56|8.34|
> |FedDyn+Default|27.84|24.89|7.39|89.59|86.65|14.56|
> |FedDyn+RandAugment|25.8|23.34|1.57|83.64|80.06|9.52|
> |FedDyn+TrivialAugment|24.5|19.7|3.81|84.34|79.06|9.4|
> |FedFA+Default|27.51|23.23|6.83|89.91|82.94|11.6|
> |FedFA+RandAugment|21.58|25.13|3.09|87.97|59.52|11.6|
> |FedFA+TrivialAugment|23.33|20.07|5.58|87.05|68.69|11.87|
> |Scaffold+Default|29.75|20.09|1.18|90.07|85.02|9.39|
> |Scaffold+RandAugment|26.56|17.06|1.18|82.57|6.52|14.45|
> |Scaffold+TrivialAugment|19.21|12.02|1.17|79.17|6.53|7.64|
> |FedAvP(local)|27.74|24.35|5.38|91.74|88.74|11.6|
> |FedAvP(Fast Update)|**31.54**|**30.96**|6.17|**92.53**|**90.13**|**18.92**|
>
> **Q1.**
>
> This question is related to W1. We conducted experiments in non-IID settings as suggested by the reviewer. We kindly ask you to refer to our response to W1. Additionally, please refer to our responses to Reviewer zQTG W1 and Reviewer WeMd W2, which include comparisons with more non-IID settings.
>
> [1] Li, Qinbin, et al. "Federated learning on non-iid data silos: An experimental study." 2022 IEEE 38th international conference on data engineering (ICDE). IEEE, 2022.

---

> > ### Comment · Reviewer_866Z · 2024-08-08
> > **Response to authors**
> >
> > I appreciate the authors' efforts on additional experiments. Given the rebuttal and other reviews, I decide to raise the rating.

---

### Official Review · Reviewer_1shA · 2024-07-09

**Soundness:** 2
**Presentation:** 2
**Contribution:** 2
**Rating:** 5
**Confidence:** 4

**Summary:**

This paper points out that the shared input-level and feature-level information poses potential privacy leakage. FedAvP only shares the augmentation policy which is not related to the data. This method leverages the first-order information as a replacement to reduce privacy leakage and communication costs. Comprehensive experiments demonstrate the efficacy and efficiency of FedAvp.

**Strengths:**

- Empirical evidence shows that the proposed method is effective in enhancing model performance.
- This method avoids the privacy leakage of the information-sharing strategy.
- This method can be deployed on different methods.

**Weaknesses:**

- The writing of this paper should be improved to express ideas clearly.
- A deeper analysis is needed to reveal why different $\alpha$ learn different augmentation methods under the same data as depicted in Figure 1. There is also no in-depth insight about how the different augmentation influences the performance in different FL setting

**Questions:**

- As FedAvp is also a learning policy for augmentation strategy. So whats the performance of FedAvg+AutoAugment which is also an automation policy?
- What is the meaning of OOD client is not mentioned.
- This paper claims that the reported accuracy is calculated as the weighted average of each client’s accuracy by the number of data points they have. However, most of the methods report the accuracy of global model. How the test dataset is partitioned of each client?

**Limitations:**

See weakness.

---

> ### Author Rebuttal · Authors · 2024-08-07
>
> **W1. Improvement to the explanation**
>
> Thank you for your valuable suggestion to improve our paper. In response, we will provide more detailed explanations to address your points, including the effect of different heterogeneity levels $\alpha$, an additional baseline, the meaning of OOD (Out-of-distribution), and the weighted average metric.
>
>
> **W2. Why does different alpha learn different augmentations?**
>
> The alpha controls the degree of data heterogeneity among clients. As the value of alpha decreases, the degree of heterogeneity increases. As the heterogeneity increases, the distribution of labels among the clients becomes more uneven, resulting in the model in each client trained from vastly different distributions even when the same dataset is used. This explains why the optimal augmentations learned by our algorithm vary with the changes in alpha. We will include figures of the dataset distributions in the appendix to further aid readers' understanding.
>
> **Q1. The performance of FedAvg + AutoAugment**
>
> The AutoAugment paper proposes a method where multiple child networks based on a controller RNN are trained through reinforcement learning. This approach was not developed with FL in consideration, where data privacy is essential. Therefore, applying AutoAugment directly to FL is challenging without further development. The original AutoAugment also requires significant computational resources, as the paper reports 5000 GPU hours for CIFAR-10 and 100 GPU hours for SVHN, making it impractical for FL environments regarding computation time and communication costs.
>
> To facilitate comparison, we applied the “pre-trained” augmentation policies reported in AutoAugment to the FedAvg algorithm. We used the CIFAR-10 and SVHN AutoAugment policies from the paper and compared the results. For CIFAR-100, we used the augmentation policy trained on CIFAR-10, as done in AutoAugment. The “Default” augmentation refers to using Random Crop and Horizontal Flip. As shown in the following table, our algorithm FedAVP shows superior performance across various heterogeneity settings for CIFAR-100, CIFAR-10, and SVHN.
>
> | Main Table               | CIFAR100/5.0 | CIFAR100/0.1 | CIFAR10/5.0 | CIFAR10/0.1 | SVHN/5.0 | SVHN/0.1 |
> |--------------------------|--------------|--------------|-------------|-------------|----------|----------|
> | FedAvg + Default         | 40.06        | 37.34        | 79.76       | 72.6        | 92.78    | 85.58    |
> | FedAvg + AutoAugment     | 47.74        | 43.45        | 81.92       | 72.12       | 92.65    | 87.02    |
> | FedAvP (Fast Update)     | 49.97        | 45.08        | 83.55       | **77.2**    | **95.14**| 87.86    |
> | FedAvP                   | **50.47**    | **45.96**    | **83.78**   | 77.1        | 95.02    | **89.81**|
>
> **Q2. The meaning of OOD client**
>
> The OOD client stands for the out-of-distribution (OOD) client, which refers to the one that does not participate in training but is used to evaluate the global model’s performance during the test. Evaluating OOD clients allows us to determine how well the FL model generalizes to new clients. Even though we briefly mentioned the OOD client in Section 4.1, the actual comparison and experimental results are exhibited in Appendix A; we will clarify this. As shown in Table 5, our approach also outperforms the experiment with the OOD clients, confirming superior generalization capacity.
>
> **Q3. The weighted average of each client’s accuracy**
>
> We followed the weighted accuracy metric according to the pFL-bench paper [1]. Based on each client's data size, the weighted average metric ensures fair representation and robustness to data imbalances, providing a more accurate overall performance evaluation. It reflects the real-world impact of larger datasets and mitigates the noise from smaller datasets, leading to a more stable assessment in FL.
>
> The label distribution of each client's test dataset is the same as the label distribution of their own training dataset. For example, in the case of the CIFAR-10 dataset, if the number of data samples per label in Client A's training dataset is [100, 30, 50, 70, 10, 80, 20, 90, 40, 60], then the number of data samples per label in Client A's test dataset will be [20, 6, 10, 14, 2, 16, 4, 18, 8, 12].
>
> Additionally, the distribution of the number of test data samples among clients is the same as the distribution of the number of training data samples among clients. Therefore, the weighted average of the global model's accuracy on each client's test dataset by the number of data samples they have is equal to the global model's accuracy on the entire test dataset.
>
>
> [1] Chen, Daoyuan, et al. "PFL-bench: A comprehensive benchmark for personalized federated learning." NeurIPS, 2022.
>
> [2] Pascanu, Razvan, et al. "On the difficulty of training recurrent neural networks." ICML, 2013.
>
> [3] Mikolov, Tomas, et al. "Empirical evaluation and combination of advanced language modeling techniques." Interspeech, 2011.
>
> [4] Shu, Jun, et al. "Meta-weight-net: Learning an explicit mapping for sample weighting." NeurIPS, 2019.
>
> [5] Zhou, Fengwei, et al. "Metaaugment: Sample-aware data augmentation policy learning." AAAI, 2021.

---

> > ### Comment · Reviewer_1shA · 2024-08-13
> >
> > Thanks to the efforts of the authors. However, I am still concerned about the insight of this paper. e.g., why FedAvP strategies tend to learn one strategy and not the other when heterogeneity is severe? This may provide new insight into addressing heterogeneity in FL. At the same time, there is a lot of work to test the global model accuracy, and the metric for this task is also needed. I  hope this paper can inspire a new view in FL to solve the heterogeneity. In summary, I will continue to maintain my grade

---

> > > ### Author Response · Authors · 2024-08-14
> > >
> > > Dear Reviewer. We sincerely appreciate your thoughtful advice.
> > >
> > > However, FedAvP does not learn just one strategy. As shown in Figure 2, it clearly learns different data augmentation strategies depending on the heterogeneity levels and distributions.
> > > As described in Section 4.3, the global policies are then adapted into a local policy using each client's local data. Figure 3 also demonstrates the statistics of personalized policies among different clients.
> > >
> > > As you suggest, we also evaluated the accuracy of the global model on CIFAR-100 ($\alpha = 5.0$ and $\alpha = 0.1$) using equally-weighted metric.
> > > We will include the results in the appendix. In this experiment, FedAvP still outperforms other baselines.
> > >
> > > |Dataset/heterogeneity degree α|CIFAR100/5.0|CIFAR100/0.1|
> > > |------------------------------|------------|------------|
> > > |FedAvg+Default|40.04|36.98|
> > > |FedAvg+RandAugment|47.3|43.17|
> > > |FedAvg+TrivialAugment|46.61| 42.04|
> > > |FedProx+Default|40.56|37.61|
> > > |FedProx+RandAugment|45.95|41.25|
> > > |FedProx+TrivialAugment|46.59|41.67|
> > > |FedDyn+Default|42.11|38.23|
> > > |FedDyn+RandAugment|45.68|42.08|
> > > |FedDyn+TrivialAugment|46.84|40.92|
> > > |FedExp+Default|42.78|38.22|
> > > |FedExp+RandAugment|46.14|41.97|
> > > |FedExp+TrivialAugment|48.54|42.01|
> > > |FedGen+Default|42.12|38.05|
> > > |FedGen+RandAugment|47.11|42.96|
> > > |FedGen+TrivialAugment|47.73|40.62|
> > > |FedMix+Default|39.59|38.46|
> > > |FedMix+RandAugment|46.67|42.7|
> > > |FedMix+TrivialAugment|46.62|42.49|
> > > |FedFA+Default|43.68|41.18|
> > > |FedFA+RandAugment|48.87|43.26|
> > > |FedFA+TrivialAugment|47.86|43.36|
> > > |FedAvP(local)|49.05|43.64|
> > > |FedAvP(Fast Update)|49.94|45.09|
> > > |FedAvP|**50.59**|**45.93**|
> > >
> > >
> > > We respectfully hope that the reviewer re-examines the evaluation of our work. Our approach to facilitating shared data augmentation policy is a novel direction in federated learning research.
> > > We have addressed data scarcity and heterogeneity, preserving security, and also provided a theoretical analysis that elucidates the role of meta-policy updates in distributed learning.

---

> > > > ### Comment · Reviewer_1shA · 2024-08-14
> > > >
> > > > Thanks to the efforts of the authors, I will raise my score.

---

### Official Review · Reviewer_WeMd · 2024-07-12

**Soundness:** 3
**Presentation:** 3
**Contribution:** 3
**Rating:** 6
**Confidence:** 3

**Summary:**

The paper introduces a novel federated data augmentation algorithm, FedAvP, designed for data augmentation of the client-side without the need to share client data information. Specifically, the authors propose a meta-learning method that allows multiple clients to collaboratively learn data augmentation policies and design a Federated Meta-Policy Loss (FMPL) for the optimization of data augmentation policies. Moreover, to prevent the gradient updates of the data augmentation policies from leaking local data privacy, they propose using first-order approximation optimization to protect privacy and reduce communication costs. Finally, the authors conducted experiments in different data heterogeneity scenarios across multiple datasets to validate the effectiveness of the method.

**Strengths:**

1) The paper innovatively applies the method of automatic data augmentation using policy loss to the federated learning scenario, to mitigate the data heterogeneity issues faced by federated learning.
2) The paper provides theoretical proof for the proposed method, and validates its effectiveness through performance experiments, privacy protection experiments, and cost experiments.
3) The paper is logically coherent and well-organized.
4) The method of multiple clients collaboratively learning automatic data augmentation policies may provide new research ideas for dealing with data heterogeneity issues in federated learning.

**Weaknesses:**

1) The augmentation policy search method in the paper only supports image data augmentation and includes only two types of augmentation methods. Moreover, as the number of augmentation methods increases, the search space will grow exponentially. The authors should analyze the scalability of FedAvP in this regard.
2) The paper lacks comparisons with the latest federated data augmentation methods, and there are few non-iid scenarios involved in the performance experiments, with only one scenario on some datasets.

**Questions:**

1. Why were the performance experiments on the Cifar-10, SVHN, and FEMNIST datasets only conducted in one scenario?
2. The gradient calculation of the policy loss introduces significant additional overhead. Could this be a hindrance to applying this method to larger visual pre-training models like ViT?

**Limitations:**

The method proposed in the paper seems to be applicable only for federated training of small-scale visual models. Its performance on tasks involving text, speech, or multi-modalities and larger-scale models still needs to be further validated.

---

> ### Author Rebuttal · Authors · 2024-08-07
>
> **W1. The scalability of FedAvP**
>
> When using two operations, the neural network output $P_{\theta}$ in the paper has a 17x17 dimension, represented as $P_{\theta}(1), ..., P_{\theta}(17 \times 17)$. From this output, we sample $P_{\theta}^{1}, ..., P_{\theta}^{B}$ based on the batch size $B$ and apply the augmentation. Since the augmentation sampling is performed from the output $P_{\theta}$ obtained through a single forward pass, even if the search space expands to 17x17x17 for three operations, the computational load of FedAvP does not increase exponentially due to only requiring one forward pass of the neural network.
> To assess whether our algorithm can effectively search within this expanded search space, we conducted the following experiment.
> We extend the layers in FedAvP to experiment with 17x17x17 possible combinations of three operations on the SVHN dataset under different heterogeneity conditions, with $\alpha = 5.0$  and $\alpha = 0.1$. The results, recorded as test accuracy per training round, are presented below.
>
> |SVHN / α=0.1|Test(%) at Round 100|Test(%) at Round 300|Test(%) at Round 500|
> |-|-|-|-|
> |FedAvP (Fast Update) / 2 layers|86.85|87.84|87.86|
> |FedAvP (Fast Update) / 3 layers|84.04|89.72|92.07|
>
> |SVHN / α = 5.0|Test(%) at Round 100|Test(%) at Round 300|Test(%) at Round 500|
> |-|-|-|-|
> |FedAvP (Fast Update) / 2 layers|92.76|94.67|95.14|
> |FedAvP (Fast Update) / 3 layers|92.73|94.44|95.01|
>
> Table: Test accuracies with different search spaces on the SVHN dataset.
>
> As shown in the table, despite the 17-fold increase in the search space when using three layers on the SVHN/0.1 dataset, FedAvP’s performance significantly improved compared to the two-layer setup reported in the main table of the paper. This demonstrates that FedAvP can effectively search the expanded search space. On the SVHN/5.0 dataset, the performance remained consistent with that of the two-layer setup. We observed that even with a 17-fold increase in the search space due to the use of three layers, FedAvP effectively searches for policies that either improve performance or maintain consistent results, depending on the data distribution.
>
> **W2. More experimental results of non-IID scenarios**
>
> We conducted additional experiments on CIFAR-10/0.1 and SVHN/5.0, which were not included in the previous paper version. The alpha in FEMNIST is not further explored due to the absence of an adjustable heterogeneity parameter. The results are provided in the official comment.
>
> **Q1.**
>
> This question is related to W2. We conducted additional experiments in non-IID environments. For a detailed explanation, we kindly ask you to refer to our response to W2. Additionally, please refer to our response to Reviewer 866Z W1, which addresses quantity-based label skew settings[7], where labels are highly partitioned.
>
> **Q2. The application of our methods to larger visual model like ViT**
>
> We reported the performance of a larger model, specifically the VGG11-s model with approximately 3x more parameters in Appendix A.2. To further investigate, we conduct experiments using the ViT-T model[1,2] on the CIFAR100/5.0 and CIFAR100/0.1, and the results are presented below.
>
> |ViT-T model|CIFAR100/5.0|CIFAR100/0.1 |
> |-|-|-|
> | FedAvg + Default       |31.75|30.71|
> | FedAvg + RandAugment   |42.39|41.7 |
> | FedAvg + TrivialAugment|41.58|33.75|
> | FedExp + Default       |37.33|35.77|
> | FedExp + RandAugment   |46.36|45.5 |
> | FedExp + TrivialAugment|44.37|40.08|
> | FedAvP (Fast Update)   |**51.1**|**47.85**|
>
> In the table, "Default" refers to the default augmentation, which applies Random Crop and Horizontal Flip. Both the FedAvg and FedExp algorithms were tested with default augmentation, as well as with RandAugment and TrivialAugment. FedAvP (Fast Update) was trained using the Fast Update method described in Section 3.3 of the paper. Our algorithm demonstrates improved performance over the CNN-based models reported in the original paper. For baseline algorithms, we observe overfitting due to the increased size of model parameters.
>
> We also compare the computation time on the client side needed to reach the target accuracy, similar to the methods reported in the paper. The results are as follows.
>
> | Computation Time of the ViT-T on CIFAR-100/5.0 | Rounds(30%) | Time(30%)    |
> |-|-|-|
> | FedAvg + Default       |400|3.08 hours|
> | FedAvg + RandAugment   |300|2.18 hours|
> | FedAvg + TrivialAugment|400|2.63 hours|
> | FedAvP (Fast Update)   |220|3.23 hours|
>
> | Computation Time of the ViT-T on CIFAR-100/0.1 | Rounds(25%) | Time(25%)    |
> |-|-|-|
> | FedAvg + Default       |300|4.33 hours|
> | FedAvg + RandAugment   |400|6.02 hours|
> | FedAvg + TrivialAugment|480|8.27 hours|
> | FedAvP (Fast Update)   |260|3.73 hours|
>
> In the first CIFAR-100/5.0 table, we recorded the number of rounds and computation time required for the global model to reach a target accuracy of 30% when trained with ViT-T model. In the CIFAR-100/0.1 table, we recorded the rounds and computation time needed to reach a target accuracy of 25%. We compared the FedAvP (Fast Update) to the FedAvg with each of the Default Augmentation, RandAugment, and TrivialAugment applied, as explained above.
> Compared to the FedAvg, FedAvP (Fast Update) takes slightly longer on the CIFAR-100/5.0 dataset. However, on the CIFAR-100/0.1 dataset, FedAvP (Fast Update) actually achieves faster learning with the ViT-T model. Overall, applying our algorithm to the ViT-T model does not result in a significant increase in computation time.
>
> [1] Dosovitskiy, Alexey, et al. "An Image is Worth 16x16 Words: Transformers for Image Recognition at Scale." ICLR, 2021.
>
> [2] Qu, Liangqiong, et al. "Rethinking architecture design for tackling data heterogeneity in federated learning." CVPR, 2022.

---

> > ### Comment · Reviewer_WeMd · 2024-08-13
> >
> > Thank you for the detailed responses. Most of my concerns have been well addressed.

---

### Official Review · Reviewer_zQTG · 2024-07-13

**Soundness:** 3
**Presentation:** 2
**Contribution:** 3
**Rating:** 6
**Confidence:** 4

**Summary:**

This paper proposes FedAvP, which performs data augmentation search by sharing policies among clients in a federated learning (FL) environment. They introduce federated meta-policy loss and utilize the first-order gradient information to further enhance privacy and reduce communication costs. The proposed algorithm allows for rapid adaptation of a personalized policy by each client, relieving the challenge of non-iid.

**Strengths:**

1. This paper proposes a novel federated data augmentation algorithm that shares only the augmentation policies during training. It allows to augment the data without revealing data privacy.

2. Although FedAvP will introduce some additional computation and communication costs, they introduce various techniques to overcome like first-order approximation and fast update.

3. The experiments are well done, and there were experimental data for all possible judging criteria. My concern is that the number of datasets is somewhat limited.

4. This paper is logically clear, and describes the possible problems and solutions one by one, which is easy to understand.

**Weaknesses:**

1. If the focus of this paper is on addressing non-iid using data augmentation, then the baselines used for comparisons are actually lacking. Classic methods for addressing non-iid, such as FedNova and Scaffold, have not been compared.
2. Additionally, to address the non-iid problem, it would be helpful to have some more vivid and intuitive explanations of why the shared policy can alleviate the non-iid issue.
3. One weakness affecting readability is that when the article references some inspired works to address certain issues, it opts for direct citations without more detailed explanations, such as gradient clipping and the reweighting in Eq. (3).

**Questions:**

1. Regarding W2, could you please add more intuitive explanations of the shared policy strategy?

2. Regarding W3, could you please explain what is the advantage of the reweighting in Eq.(3)?

**Limitations:**

The authors provide discussions on the limitations of the work in Sections 5.

---

> ### Author Rebuttal · Authors · 2024-08-07
>
> **W1. Comparison with other classic non-IID methods, such as FedNova and Scaffold.**
>
> In Table 1 of the manuscript, we compare our model with baselines, including state-of-the-art federated learning and federated data augmentation algorithms. As the review suggested, we conducted an additional experiment with non-IID algorithms, including FedNova [1] and Scaffold [2].
>
> |Dataset / heterogeneity degree α|CIFAR100/0.1|CIFAR10/0.1|SVHN/0.1|FEMNIST|
> |-|-|-|-|-|
> |FedNova + Default        |38.52|74.45|88.16|81.21|
> |FedNova + RandAugment    |42.43|74.08|84.42|79.79|
> |FedNova + TrivialAugment |40.23|71.99|82.96|78.92|
> |Scaffold + Default       |44.94|75.67|87.26|83.17|
> |Scaffold + RandAugment   |43.57|72.4|77.07|79.31|
> |Scaffold + TrivialAugment|42.14|64.12|14.70|78.06|
> |FedAvP(Fast Update)             |45.08|**77.2**|87.86|**84.47**|
> |FedAvP                   | **45.96**|77.1|**89.81**|84.27|
>
> "Default" refers to the default augmentation, which applies Random Crop and Horizontal Flip. Both the FedNova and Scaffold algorithms were tested with default augmentation, as well as with RandAugment and TrivialAugment. FedAvP (Fast Update) was trained using the Fast Update method described in Section 3.3 of the paper. We examined the performance under diverse heterogeneity conditions on highly non-IID environments using CIFAR100, CIFAR10, SVHN, and FEMNIST datasets. Our FedAvP method outperformed across all tested scenarios.
>
>
>
> **W2. Why the shared policy can alleviate the non-iid?**
>
> When the client's data distribution follows non-i.i.d. in federated learning, local models are trained on the different data distributions, which can lead to poor global model aggregation due to distribution shifts. However, our shared data-augmentation policy and meta-policy search strategy can generate additional data samples that help balance the data across clients, thereby reducing the disparity and scarcity in data distributions and improving model convergence and accuracy. As shown in Figure 1(a), under the 'local policy update,' our algorithm updates the augmentation policy so that the aggregated model performs well across the various local data distributions (non-iid, pre-augmentation).
>
> **W3. The detailed explanations of gradient clipping and reweighting**
>
> We sincerely thank the reviewer for the suggestion regarding the improvement of the readability of our paper. Gradient clipping [3,4] is performed using the method $g \leftarrow \frac{c}{||g||}$ when $||g|| > c$, and it is applied to the gradients of $D_{k}^{val}$ and $D_{k,n-1}^{train}$ in Eq. (6). Reweighting [5,6] is performed using the sigmoid output of the neural network $P_{\theta}$ in Eq. (3). Specifically, when the search space is a 17x17-dimensional neural network output $P_{\theta}$, each dimension $P_{\theta}(1), ..., P_{\theta}(17 \times 17)$ corresponds to a two-operations augmentation, where a higher value indicates a higher probability of selecting that augmentation. During training, when there is a batch size of $B$, $B$ samples are drawn from a multinomial distribution based on the value of $P_{\theta}$, resulting in $P_{\theta}^1,..., P_{\theta}^B$. Here, each $P_{\theta}^i$ corresponds to one of the 17x17 neural network output dimensions.
> For each data sample, two-operations augmentation is applied based on $P_{\theta}^i$, and the weight of the augmented data sample is determined by the corresponding $P_{\theta}^i$. We will include this explanation in the Appendix.
>
> **Q1.**
>
> Please refer to our response to W2.
>
> **Q2. The advantage of the reweighting in Eq.(3).**
>
> We applied the corresponding two-operations to each data sample in the batch using the 17x17-dimensional neural network output, where the weight of the augmented sample was determined by $P_{\theta}^i$.  After training the local model, our policy loss tries to improve the performance on the validation data.
> In this process, two-operations that hinder the performance on the validation data are updated through backpropagation using the weight $P_{\theta}^i$, causing the weight $P_{\theta}^i$ to decrease.
>
> Our Federated Meta-policy (FMPL) approach updates a shared data augmentation policy for each client’s unique environment to mitigate data heterogeneity. The sampling operation is non-differentiable in the typical data-augmentation approach. However, the reweighting approach in Eq.(3) enables differentiation, which is a key advantage.
>
>
> [1] Wang, Jianyu, et al. “Tackling the objective inconsistency problem in heterogeneous federated optimization.” NeurIPS, 2020.
>
> [2] Karimireddy, Sai Praneeth, et al. “Scaffold: Stochastic controlled averaging for federated learning.” ICML, 2020.
>
> [3] Pascanu, Razvan, et al. “On the difficulty of training recurrent neural networks.” ICML, 2013.
>
> [4] Mikolov, Tomas, et al. “Empirical evaluation and combination of advanced language modeling techniques.” Interspeech, 2011.
>
> [5] Shu, Jun, et al. “Meta-weight-net: Learning an explicit mapping for sample weighting.” NeurIPS, 2019.
>
> [6] Zhou, Fengwei, et al. “Metaaugment: Sample-aware data augmentation policy learning.” AAAI, 2021.
>
> [7] Li, Qinbin, et al. “Federated learning on non-iid data silos: An experimental study.” ICDE. IEEE, 2022.

---

> > ### Comment · Reviewer_zQTG · 2024-08-12
> > **Thank you**
> >
> > Thank you for the responses. Most of my concerns have been addressed.

---

### Author Rebuttal · Authors · 2024-08-07

**More experimental results**
We conducted additional experiments to answer Reviewer WeMd's W2 and Reviewer 866Z's W1, considering more non-IID settings and highly partitioned label skew settings [1], respectively.

|Dataset/heterogeneity degree α|CIFAR100/5.0|CIFAR100/0.1|CIFAR10/5.0|CIFAR10/0.1|SVHN/5.0|SVHN/0.1|FEMNIST|
|------------------------------|------------|------------|-----------|-----------|--------|--------|-------|
|FedAvg+Default|40.06|37.34|79.76|72.6|92.78|85.58|80.65|
|FedAvg+RandAugment|47.29|43.6|82.82|73.73|92.48|84.84|79.4|
|FedAvg+TrivialAugment|46.61|42.16|82|71.09|91.99|83.36|79.01|
|FedProx+Default|40.57|37.71|80.64|73.23|93.15|86.79|81.45|
|FedProx+RandAugment|45.97|41.39|82.56|73.71|92.33|85.52|77.11|
|FedProx+TrivialAugment|46.61|41.81|81.83|70.89|91.67|84.11|79.67|
|FedDyn+Default|42.09|38.52|80.36|73.86|93.16|87.6|80.47|
|FedDyn+RandAugment|45.7|42.24|82.51|72.78|92.16|81.47|77.64|
|FedDyn+TrivialAugment|46.83|41.1|82.03|70.34|92.22|83.41|79.31|
|FedExp+Default|42.76|38.28|80.64|73.7|92.77|86.66|81.45|
|FedExp+RandAugment|46.13|42.23|82.86|70.78|92.12|84.63|79.69|
|FedExp+TrivialAugment|48.55|42.09|82.51|71.07|92.64|83.72|80.2|
|FedGen+Default|42.14|38.27|80.23|72.74|92.71|86.79|81.86|
|FedGen+RandAugment|47.11|43.1|81.9|73.42|91.84|84.39|79.34|
|FedGen+TrivialAugment|47.71|40.76|82.58|70.87|91.73|83.23|77.35|
|FedMix+Default|40.26|38.69|80.99|74.54|92.8|86.02|81.63|
|FedMix+RandAugment|46.69|43|83.08|74.25|92.36|83.44|79.46|
|FedMix+TrivialAugment|46.64|42.63|81.83|71.5|91.85|82.34|77.84|
|FedFA+Default|43.7|41.21|82.61|76.02|92.77|87.33|81.13|
|FedFA+RandAugment|48.86|43.44|82.44|73.53|91.21|81.32|78.71|
|FedFA+TrivialAugment|47.86|43.45|80.12|72.89|91.89|78.62|78.96|
|FedAvP(local)|49.04|43.86|83.64|73.43|94.71|87.05|83.94|
|FedAvP(Fast Update)|49.97|45.08|83.55|**77.2**|**95.14**|87.86|**84.47**|
|FedAvP|**50.47**|**45.96**|**83.78**|77.1|95.02|**89.81**|84.27|

Our algorithm achieves the highest performance across all scenarios.

[1] Li, Qinbin, et al. “Federated learning on non-iid data silos: An experimental study.” ICDE. IEEE, 2022.

---

### Decision · Program_Chairs · 2024-09-25

**Decision:**

Accept (poster)

**Comment:**

This paper proposes FedAvP, a data augmentation methods by sharing policies among clients in a federated learning environment. It focuses on an important and challenging aspect in federated learning. The effectiveness of the proposed solution is verified by extensive experiments. We recommend accept it.